# DLK orchestrates a modular transcriptional response to axon injury with separate roles for Fos and Jun

Gibarni Mahata, Li Chen, Gregory O. Kothe, Melissa M. Rolls *

Biochemistry and Molecular Biology and the Huck Institutes of the Life Sciences, The Pennsylvania State University, University Park, Pennsylvania, United States of America

* mur22@psu.edu

## Abstract

Axon injury initiates transcriptional reprogramming that in competent cells leads to regeneration. In vertebrate neurons, DLK acts upstream of Jun, STAT and Atf3, core transcription factors that mediate regeneration. It is unclear whether these three proteins are activated independently, or whether they function in a linear cascade. To investigate relationships between these transcription factors we wished to use Drosophila as a model system as it has one ortholog of each. However, the only transcription factor linked to DLK-mediated axon regeneration (AR) in flies was Fos. Using loss of function approaches we demonstrate that Jun, STAT and Atf3 are required for Drosophila sensory axon regeneration, indicating transcriptional control of axon regeneration is broadly conserved. We next investigated temporal roles for Fos, Jun, STAT and Atf3. Only Fos is required for the early transcriptional response, which coincides with neuroprotection, and its nuclear entry and homodimerization coincide with this phase. Reduction of Jun homodimerization occurs after axon injury downstream of DLK/JNK, but independently from Fos, at a later stage associated with axon regrowth. STAT nuclear entry occurs downstream of Jun as part of this stage, is inhibited by Fos, and does not require JAK, which is dispensable for axon regeneration. Atf3 nuclear exit is in turn downstream of Fos, Jun, and STAT. Our results suggest that DLK/JNK separately activates Fos and Jun, and that Jun initiates a transcriptional cascade that includes STAT and Atf3. These two transcriptional modules control separate steps of the injury response that culminates in axon regeneration.

## Author summary

Neuronal cells, unlike other cell types, cannot divide or be replaced after injury. Axon damage is particularly deleterious as many neurons have a single axon that is required to relay signals to target cells. Axon regeneration allows some

**Data availability statement:** All relevant data are in the manuscript and its supporting information files.

**Funding:** Funding for this work was provided in part by the National Institute of Neurological Disorders and Stroke R01 NS121245 to MMR (https://www.ninds.nih.gov/). The funder had no role in the design of the study, data collection and analysis, decision to publish or preparation of the manuscript.

**Competing interests:** The authors have declared that no competing interests exist.

neurons to recover from axon injury. DLK is a critical sensor of axon damage and it activates downstream signaling proteins to launch injury responses. DLK engages multiple transcription factors after axon injury and it is unclear if they act independently or as part of a connected sequence. We show that DLK-regulated transcription factors- Fos, Jun, STAT and Atf3 are required for axon regeneration in a simple Drosophila model, but act during different phases. We find that Fos is required for early axon injury responses that stabilizes the cell. Jun is activated later by DLK, independent of Fos and it regulates STAT and Atf3. This second module is needed for later injury responses including axon regeneration. Our study highlights how these proteins are organized over time to coordinate distinct phases of axon injury response enabling neuronal survival and regeneration after injury.

## Introduction

Many neurons possess a single axon that extends far from the cell body and encounters complex environments. If the axon is damaged, the cell body must be notified so that an appropriate response can be executed. In the vertebrate peripheral nervous system (PNS) the axon regeneration program is activated and this can allow neurons to regain function [1–4]. Initiating axon regeneration involves chromatin reprogramming and transcription of hundreds to thousands of regeneration associated genes (RAGs) [5–8]. While several types of signals including a wave of calcium [1,3,9,10] reach the cell body after an axon is injured to alert it to the distant damage, dual leucine zipper kinase (DLK) is essential to injury sensing in many neuron types. DLK, a conserved MAP kinase kinase kinase, is activated by axon damage in *C. elegans*, Drosophila, and vertebrates and is required to initiate injury responses including regeneration in these animals [11–16]. In each of these systems, DLK has been linked to one or more downstream transcription factors, but the relationships between these transcription factors have not been untangled. DLK could activate each independently, or their activation could be coordinated in a cascade that sequentially turns on steps in the injury response. One difficulty in establishing the relationships between injury-activated transcription factors is that they have been studied in different systems, so another possibility is that DLK activates different sets of transcription factors in different species or cell types to initiate axon regeneration.

DLK was first linked to axon injury responses in *C. elegans* [11,17] where it is essential for regeneration of motor [11] and sensory axons [17]. In this system, the bZip transcription factor cebp-1 (the *C. elegans* C/EBP) acts downstream of DLK to promote axon regeneration [10,17]. DLK was next shown to be critical for axon injury responses in Drosophila motor neurons [15] where it acts through the AP-1 bZip transcription factor Fos to initiate injury-induced transcription [15]. In mammalian neurons DLK acts upstream of many RAGs [18] and several different transcription factors have been linked to DLK signaling after axon injury in mammals. cJun, an AP-1 bZip transcription factor important for axon regeneration of the mouse facial nerve [19,20]

and expression of many RAGs [21], fails to accumulate in the phosphorylated form in nuclei after sciatic nerve injury [18] without DLK. Similarly, STAT3, which is important for regeneration of sensory axons in mice [22], is not phosphorylated and concentrated in the nucleus in the absence of DLK [14]. Atf3, a bZip transcription factor that promotes facial nerve regeneration [23] and enhances regenerative growth of axons in the sciatic nerve when overexpressed [24] requires DLK to increase in nuclei after retinal ganglion cell axon injury [16]. Thus, these five transcription factors, Fos, Jun, C/EBP, Atf3 and STAT3 have all been shown to be regulated by DLK after axon injury, and their involvement in axon injury responses is supported by functional data. We therefore wished to focus on this set as a starting point for understanding the relationships between DLK-regulated transcription factors.

Several different hypotheses about how transcription factors might be regulated by DLK after axon injury are possible. At one extreme, each transcription factor could be activated independently by DLK. For example, each could be directly phosphorylated by DLK or its downstream MAP kinase c-Jun N-terminal kinase (JNK). At the other extreme, only one transcription factor might be directly regulated by DLK/JNK, and the rest could be activated in a linear cascade downstream of this transcription factor. In between these extremes, subsets of the core regeneration transcription factors might work together. For example, Fos and Jun are both bZip transcription factors that bind AP-1 sites and canonically work together [25,26], but Jun can also bind Atf3 to promote neurite sprouting in neuron-like cells [27]. To try to understand how transcription factors are coordinated downstream of DLK, we needed a system in which multiple transcription factors are required, where tools to address relationships between these transcription factors exist, and where distinct steps in the injury response can be used to establish the temporal sequence of transcription factor function.

Axon regeneration in Drosophila sensory neurons has features that make it promising for investigating relationships between transcription factors in the DLK injury response. First, it depends absolutely on DLK; after the axon of sensory neurons is severed, no regeneration is initiated in *DLK* mutant animals [28]. Second, several different steps can be identified after axons are injured. Within 8 hours after axon injury microtubule dynamics is dramatically upregulated in the cell body and dendrites and this response peaks by 24 hours before subsiding by 48 hours [29]. Outgrowth of the regenerating axon typically begins 24–48 hours after injury [30]. The early microtubule dynamics response to axon injury is neuroprotective and serves to stabilize the cell [29] and requires DLK, JNK and Fos [31]. It is turned down by 48 hours after axon injury by a pathway that involves mitochondrial fission and caspases [31]. If microtubule dynamics and neuroprotection are forced to remain high by caspase reduction or overexpression of Fos then axon regeneration is reduced [31]. Thus, the early neuroprotective phase is a step that must be turned down for optimal regeneration. Third, axon regeneration can be easily quantitated in individual cells after laser-mediated proximal axon severing. Cutting axons close to the cell body promotes regeneration by converting a dendrite into a new axon rather than growth from the stump in Drosophila [30,32] and mammalian [33] neurons. In Drosophila sensory neurons, the injury signaling pathway is the same whether growth initiates from a dendrite or the axon stump [32], but regeneration from a dendrite confines the new axon to the body wall in most cases where it can be easily measured (Fig 1A and 1B). Finally, each of the five DLK-dependent transcription factors noted above has a single ortholog in Drosophila. In addition to these useful features for assessing transcription factor relationships in the axon injury response, there are some potential drawbacks. Most notably, only Fos has been shown to be activated by DLK in Drosophila [15] and while it has been linked to the early neuroprotective phase of the injury response in sensory neurons [31], its role in sensory axon regeneration has not been established.

To determine whether we could use Drosophila sensory neurons as a model system to understand the DLK-dependent transcriptional response to axon injury, we first tested the Drosophila orthologs of the five transcription factors described above: Fos (kay in flies), Jun (jra in flies), STAT (Stat92E in flies), Atf3 and c/ebp (slbo in flies). Reduction of four of these proteins strongly reduced axon regeneration, so we used a variety of approaches to understand their temporal dynamics and regulatory interactions. Using genetic approaches and live imaging *in vivo*, we found that Fos, STAT and Atf3 exhibit distinct axon injury-induced temporal profiles of nuclear accumulation. In line with early nuclear accumulation, only Fos is required for the early transcriptional response. Consistent with different temporal requirements for Fos and Jun, they

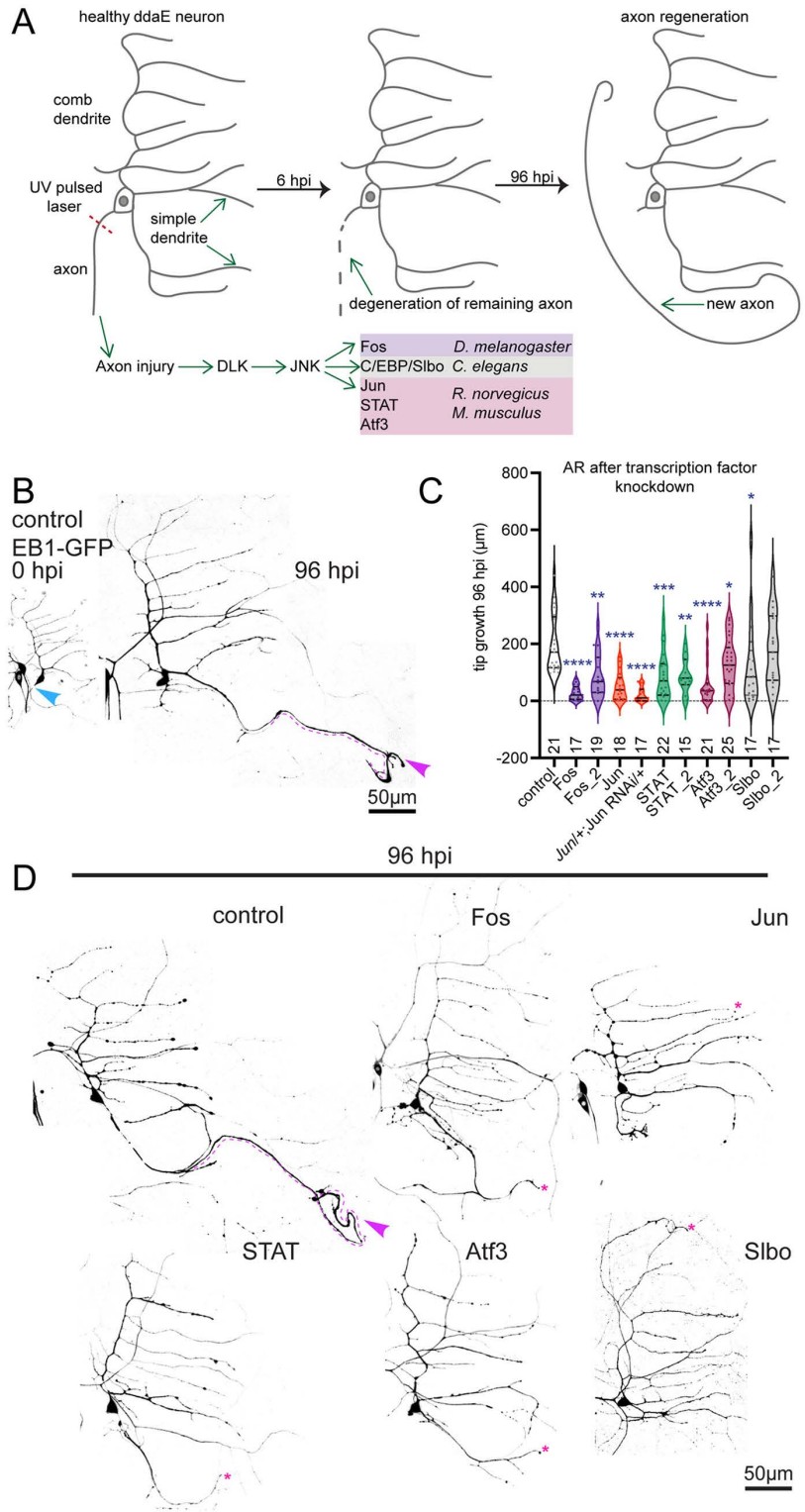

**Fig 1. Axon regeneration in Drosophila Class I (ddaE) sensory neurons requires a conserved suite of transcription factors.** (A) Schematic representation of axon regeneration (AR) assay in Drosophila Class I ddaE neurons. Axotomy is performed proximal to the cell body resulting in degeneration of the severed axon and subsequent conversion of one dendrite into a regenerating axon. Axon injury is sensed by DLK which activates

a JNK-mediated signaling cascade. This pathway is transcriptionally regulated by various transcription factors across model organisms. (B) Representative images of ddaE neurons taken at 0 hpi and 96 hpi. Blue arrow marks cut site and purple dashed line and arrow indicate the regenerated axon. (C) Quantification of new axon tip growth length at 96 hpi for each genetic condition. RNAi-mediated knock down of Fos, Jun, Atf3, STAT, Slbo and a trans-heterozygous combination of *Jun* mutant allele with Jun RNAi significantly reduce axon tip growth length compared to the control. The numbers above each genotype are the numbers of animals tested for that condition. Genotypes are listed in S1 Table. Fos and Fos_2 are two independent RNAi transgenic lines (BDSC 27722 and BDSC 33379), as are STAT and STAT_2 (VDRC 106980 and BDSC 31317), Atf3 and Atf3_2 (VDRC 105036 and BDSC 26841) and Slbo and Slbo_2 (BDSC 53309 and BDSC 27043). The Jun RNAi line here is BDSC 31595. To generate heterozygous *Jun* mutant animals with Jun RNAi, the Jun RNAi was introduced into the background of the *Jun* null mutant *Jra[IA109]*. (D) Example images of each genotype at 96 hpi are shown. The dendrite that took on axonal microtubule polarity, but then failed to grow, is indicated with a pink star. The one exception to this is Fos RNAi where polarity does not reverse. In this case the dendrite closest to the axon was used for measurement as this one becomes the new axon most frequently in control neurons [30]. For statistical analysis, Kruskal-Wallis one way analysis of variance (ANOVA) was used. In the plot, the thick line represents the median, while the dashed lines represent first and third quartiles. * $p < 0.05$, ** $p < 0.01$, *** $p < 0.001$, **** $p < 0.0001$.

do not heterodimerize after axon injury. Instead, Fos homodimerizes with itself while Jun switches its binding partner post-axotomy. Using nuclear levels of STAT and Atf3 as readouts, we show that STAT nuclear entry requires Jun, and Atf3 nuclear dynamics is downstream of Fos, Jun and STAT. In summary, we show that DLK activates a temporally dynamic transcriptional program for both early and later axon injury responses. Fos drives early axon injury responses associated with neuroprotection, independent of its canonical binding partner, Jun. At later stages, DLK/JNK signaling facilitates a switch in Jun binding partners. STAT nuclear entry occurs downstream of Jun and likely contributes to the later regenerative phase including axon outgrowth. Consistent with different roles for Fos and Jun, Fos inhibits nuclear entry of STAT. These upstream DLK-dependent transcriptional programs ultimately converge on Atf3 regulation, which exhibits a biphasic nuclear localization. The final model is a hybrid one in which Fos and Jun are activated independently by DLK signaling at different times after injury, but STAT and Atf3 act as part of a secondary network downstream of Fos and Jun.

## Results

### Multiple transcription factors are required for regeneration of Drosophila sensory axons

In Drosophila sensory neurons when axons are severed near the cell body (within ~30 μm, termed proximal axotomy) one of the dendritic branches converts into a regenerating axon, acquiring plus-end-out microtubule polarity and molecular characteristics of an axon [30,32]. This conversion has been described in Class IV sensory neurons [32], which function in larvae as nociceptors to detect parasitic wasps [34], and Class I sensory neurons [30], which inform the central nervous system about direction of larval locomotion [35,36]. We used the ddaE Class I sensory neuron as a model neuron for this study as it is easy to measure the length of the new axon after proximal laser-mediated severing (Fig 1A and 1B). To determine which of the transcription factors previously shown to be 1) activated by DLK, and 2) involved in axon injury responses (Fig 1A) is required for regeneration in this cell, we used RNA interference (RNAi) to deplete their Drosophila orthologs. RNAi hairpins as well as cell shape markers were expressed using the Gal4-UAS binary system [37] so that only the cells we were visualizing experienced knockdown, while the rest of the animal was wild-type.

We performed proximal axotomy on Class I ddaE neurons in intact larvae using a UV-pulsed laser and measured growth of the newly specified axon length 96 hours post-injury (hpi) (Fig 1B). As expected, robust regeneration was observed in the control post axotomy where γTub37C, a maternal γTubulin is not expressed in somatic cells [38], was targeted by RNAi. Knockdown of Fos, Jun, Atf3, STAT, and Slbo significantly reduced AR compared to the control (Fig 1C and 1D). To confirm the robustness of these phenotypes, we tested other RNAi lines available for Fos, STAT, Atf3 and Slbo, and combined the Jun RNAi with one copy of *Jun* mutant and observed a similar decrease in axon regeneration for Fos, Jun, STAT and Atf3 (Figs 1C, S1A and S1B). Some STAT RNAi lines did not show an axon regeneration phenotype, so we assessed their knockdown efficiency by quantifying the nuclear fluorescence intensity of GFP-tagged STAT expressed under its endogenous regulatory region (https://flybase.org/reports/FBal0268865). We observed that the RNAi

lines that failed to reduce axon regeneration had a weaker reduction in basal nuclear levels of STAT-GFP levels (S1A and S1B Fig).

Together, these data demonstrate that five transcription factors most closely linked to DLK signaling after axon injury across multiple systems are all required for axon regeneration in Drosophila making it a useful system to understand their relationships. Among them, Slbo showed the weakest axon regeneration phenotype, so we chose to pursue the other four transcription factors (Fos, Jun, Atf3, STAT) for additional experiments.

## Fos controls early injury responses without Jun

To begin to tease apart when the four transcription factors we selected function in the axon injury response, we used a puc-GFP protein trap [39] to report on early transcriptional changes downstream of DLK. Puc is a MAP kinase phosphatase that acts as a negative feedback modulator of JNK signaling [40,41]. In Drosophila motor neurons, a puc transcriptional reporter was shown to be upregulated by axon injury and this was blocked by loss of DLK or expression of a dominant negative (bZip domain only) form of Fos, but not a similar form of Jun [15]. In Drosophila sensory neurons, puc-GFP has been shown to accumulate in the nucleus after axon injury with about a two-fold increase 6h after injury and 4-fold increase 24h after injury [28]. The time course of puc-GFP nuclear accumulation is thus similar to the timing of increased microtubule dynamics and neuroprotection [29] and thus serves as a marker of this early phase of injury responses.

To quantitate puc-GFP nuclear accumulation after injury and in different genetic backgrounds we compared nuclear fluorescence in two sister sensory neurons. We injured the axon of the ddaE neuron while sparing its neighboring ddaD neuron (Fig 2A). In control RNAi neurons, puc-GFP levels were about 1.8-fold higher in the nucleus of the injured ddaE cell compared to uninjured ddaD at 6 hours after axon severing (Fig 2B) consistent with our earlier studies [28]. RNAi reduction of DLK or Fos blocked the injury-induced nuclear increase of puc-GFP (Fig 2A and 2B). Knockdown of Jun had no effect on nuclear puc-GFP intensity (Fig 2A and 2B). Given that RNAi-mediated knockdown may cause incomplete suppression of Jun expression, we further reduced Jun by pairing the RNAi transgene with one copy of a *Jun* mutant. This combination had a stronger effect on axon regeneration than Jun RNAi alone (Fig 1C) but did not block the increase in nuclear puc-GFP at 6 hpi (Fig 2B). Similarly, reduction of STAT or Atf3 by RNAi did not affect the induction of puc-GFP by axon injury (Fig 2A and 2B). For this experiment, we used RNAi fly lines that exhibited strongest reduction of axon regeneration and high knockdown efficiency (Figs 1 and S1A–S1C). These results suggest that, of these four transcription factors, only Fos acts in the early step of the axon injury response. It was particularly surprising that Jun was not required with Fos, as these two are thought to act together as the heterodimeric AP-1 transcription factor. To further confirm that they act separately after axon injury, we examined the effect of Fos and Jun overexpression.

A previous study showed that Fos overexpression reduces axon regeneration in ddaE, likely due to its role in neuro-protection, which needs to be downregulated to allow axon regeneration to proceed [31]. If Jun works together with Fos to mediate the early neuroprotective injury response, then overexpression of Jun might also reduce regeneration. We confirmed that Fos overexpression reduces axon regeneration as reported (Fig 2C and 2D) but found that two different Jun overexpression transgenes did not alter axon regeneration (Fig 2C and 2D). Moreover, co-overexpression of Fos and Jun suppressed axon regeneration to a similar extent as overexpression of Fos alone (S1D and S1E Fig). Thus, loss and gain-of-function approaches support a role for Fos in early axon injury responses that is independent from Jun.

## Fos levels increase in the nucleus after axon injury, while Jun remains constant

To better understand the temporal regulation of Fos and Jun after axon injury, we wished to track their localization. We therefore generated transgenic flies in which Fos and Jun were tagged at their endogenous loci with the mNeonGreen (mNG) coding sequence (Fig 3A) and examined their localization after axon injury. As Fos and Jun are both essential genes, to test whether tagging disrupted function we generated animals in which the only copies of Fos and Jun were the mNG tagged ones. It was possible to generate homozygous viable stocks with both Fos-mNG and Jun-mNG indicating

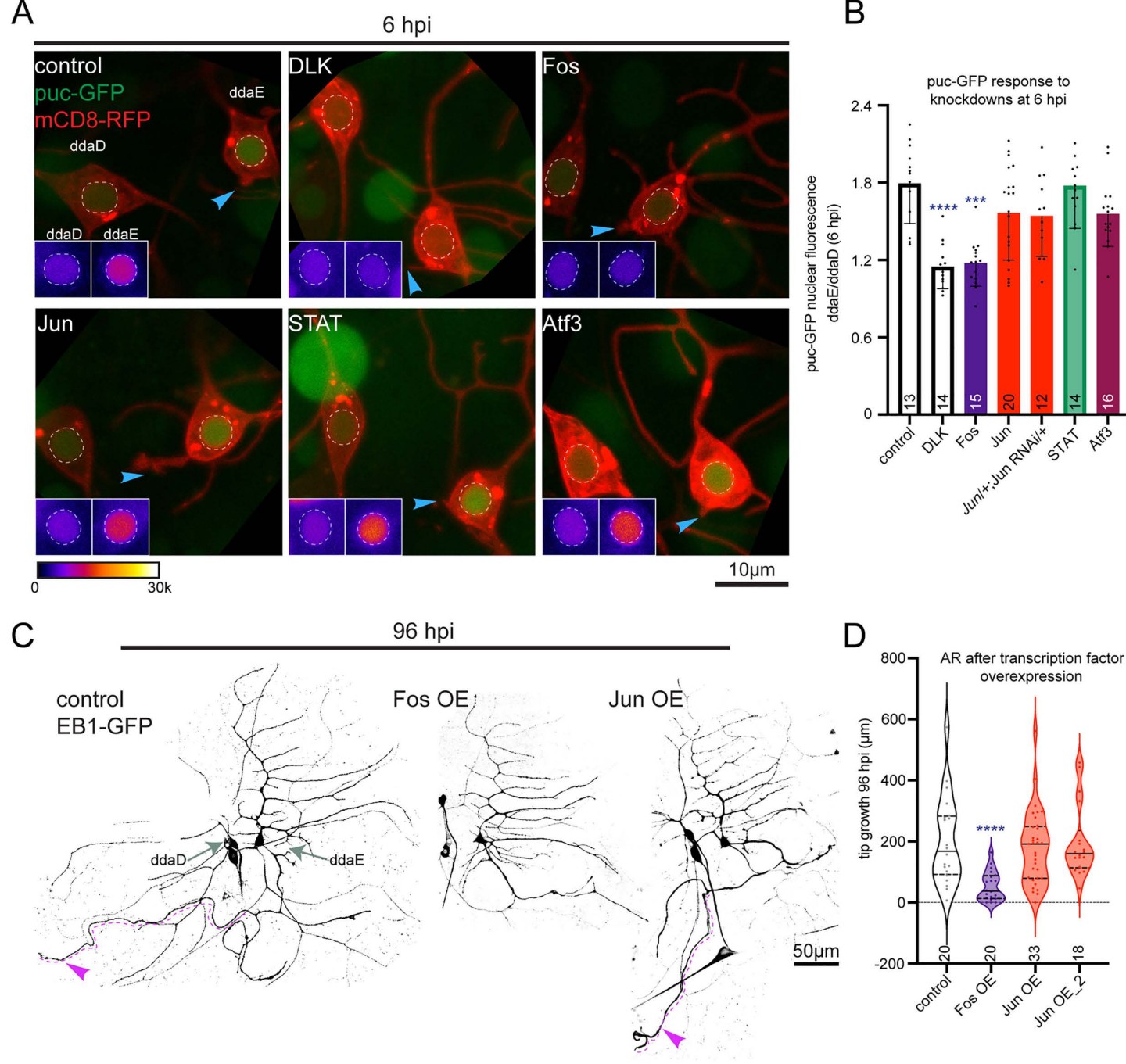

**Fig 2. Fos is required for early injury responses, whereas Jun is not.** Axons in ddaE neurons were severed near the cell body and images were acquired to compare nuclear fluorescence intensity of puc-GFP in injured ddaE versus adjacent uninjured ddaD cells with different genetic backgrounds. In all panels and insets ddaE is to the right and ddaD is to the left. (A) Example images showing nuclear localization of puc-GFP at 6 hpi. Blue arrow represents the cut site. Insets show the nuclear region of interest (marked with dashed circle) visualized using a fire lookup table (LUT), where white represents the highest fluorescence intensity and black represents lowest. (B) Quantification of the ratio of puc-GFP fluorescence intensity in injured ddaE to uninjured ddaD in DLK, Fos, Jun, Atf3, STAT knockdown and combination of *Jun* mutant allele with Jun RNAi backgrounds, where only DLK and Fos knockdown block the increase of nuclear puc-GFP post axon injury. (C) Fos and Jun were overexpressed in Class I sensory neurons and new axon tip growth length was measured at 96 hpi. Two different UAS-Jun transgenes were used. Representative images show that overexpression of Fos reduced

tip growth length at 96 hpi while Jun overexpression had no effect on AR. Purple arrows mark regenerated axon. (D) Quantification of axon tip growth length for each overexpression line. Numbers above genotypes are numbers of animals used in each condition. Statistical significance was calculated using Kruskal-Wallis one way ANOVA. Error bars represent SD (B), and the thick line and the dashed lines represent median, and first and third quartiles respectively (D). ***p < 0.001, ****p < 0.0001.

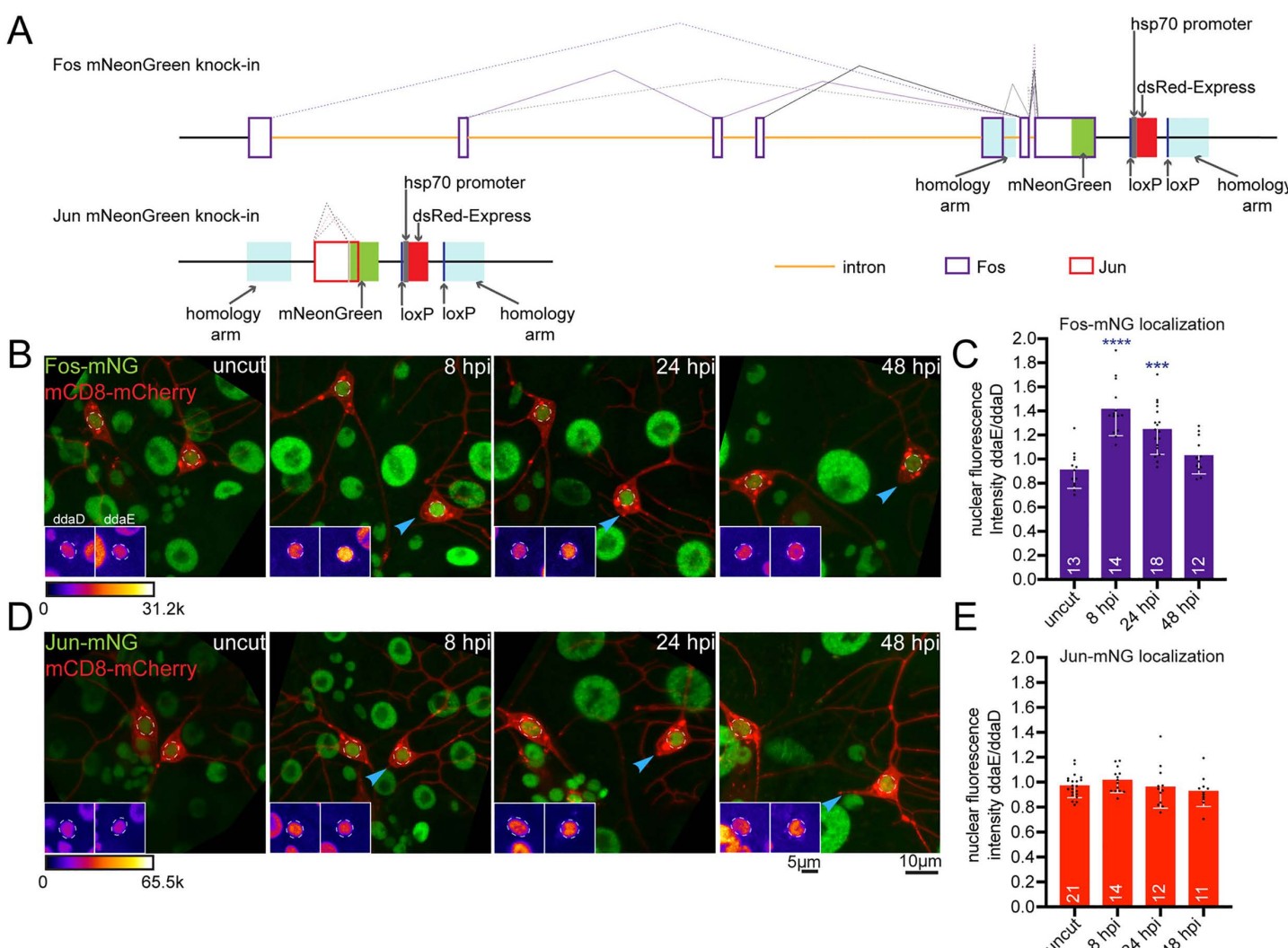

**Fig 3. Nuclear localization of Fos is dynamically regulated post axon injury.** Endogenous Fos and Jun loci were tagged with mNeonGreen (mNG) using CRISPR/Cas9 to monitor nuclear localization in class I sensory neurons at the basal state and 8, 24, 48 hpi. (A) Schematic representation showing the Fos-mNG and Jun-mNG knock-in constructs generated using CRISPR/Cas9. (B) Representative images of Fos-mNG at indicated time points post axon injury. Nuclear fluorescence intensity increases by 8 hpi and gradually declines 24-48 hpi. ddaE (right) was injured and ddaD was not (left). Insets show FIRE LUT of the Fos channel. (D) Example images of Jun-mNG before and after axon cut (different timepoints post-injury), showing relatively stable nuclear localization over time. In both (B) and (D), blue arrows represent the cut site. Insets show the Jun channel in the region of interest indicated by dashed circle, visualized using fire LUT. (C and E) Quantification of nuclear fluorescence intensity of Fos-mNG (B) and Jun-mNG (D) in ddaE (cut), normalized to ddaD (uncut) neuron. All the statistical analyses in (C) and (E) were performed using Kruskal-Wallis one way ANOVA test and the error bars in the graphs represent SD. Numbers in the bars are the number of animals tested for that condition. ***p < 0.001, ****p < 0.0001.

that the tag does not disrupt normal function. Consistent with an early role for Fos, nuclear fluorescence of endogenous Fos-mNG was higher in injured ddaE neurons than uninjured ddaD neurons at 8 hpi (Fig 3B and 3C). However, by 24 hpi, the fluorescence intensity began to decline and was further reduced at 48 hpi (Fig 3B and 3C). We validated these results using a second data set (S2A Fig). This pattern is consistent with the requirement of Fos for neuroprotection, which peaks at 24 hpi and is complete by 48 hpi [29], as well as the inhibitory effect of Fos overexpression on the later outgrowth phase (Fig 2D). As expected, injury-induced nuclear entry of Fos-mNG depended on DLK (S2B and S2C Fig). The simplest model for Fos function is that it acts early and independently from other transcription factors to control neuro-protection, which is a response that competes with axon regeneration, and the others act later to separately control axon outgrowth.

In contrast to Fos, the fluorescence intensity of endogenous Jun-mNG in the nucleus was not increased by injury (Fig 3D and 3E), although overall levels of nuclear Jun increased in the uninjured ddaD and injured ddaE neurons over time (Fig 3D). This result is consistent with analysis of Jun nuclear localization in mammalian cells, where it does not depend on phosphorylation by JNK or dimerization [42]. While nuclear localization was useful for Fos to confirm its time of action, it did not provide insight for Jun or help identify what each might partner with if they were not working together.

## Fos homodimerization is stimulated by axon injury, while Jun-Jun and Fos-Jun interactions are reduced

The requirement for Fos but not Jun for early injury responses suggested that these two bZip transcription factors do not work together after axon injury. However, bZip transcription factors dimerize when they bind DNA [43,44] so if Fos and Jun do not work together, they must either form homodimers or dimerize with a different bZip transcription factor. To identify binding interactions between dimeric transcription factors in injured cells we chose to use bimolecular fluores-cence complementation (BiFC) as binding interactions can be easily detected in living tissue by increases in fluorescence signal [45]. In this assay, GFP-derived fluorescent proteins are split into two pieces that do not have high enough affinity to interact and form a complete fluorophore on their own. However, when binding partners are attached to the two parts, their interaction can bring together the fluorescent protein pieces and allow them to assemble and become fluorescent [45]. This assay was originally developed to monitor interactions between bZip transcription factors [46] and has subse-quently been used to map many types of interactions, including a large-scale project to detect transcription factor interac-tions in Drosophila [47]. It was initially proposed that once complete fluorescent proteins formed, the interactions would be irreversible [46], which would hinder assessment of dynamic interactions. However, subsequent studies have challenged this notion and shown convincing evidence of rapid (minute) timescale reversal of BiFC signal [48,49]. We used Venus (a YFP-derivative) based BiFC so that we could take advantage of existing Drosophila tools [47]. The N-terminal part of Venus (VN) can reconstitute a YFP-like fluorescent protein with either the C-terminal part of Venus (VC) or Cerulean (CC) (Fig 4A). We generated BiFC Fos-VC and -VN constructs and transgenic flies (Fig 4B) and used available Jun-VN [50], Fos-CC and Jun-CC [47] transgenic strains (Fig 4A and 4B). We used the Gal4-UAS system to express these tagged pro-teins in class I sensory neurons in different combinations. At the basal uninjured state, we observed nuclear BiFC signals for Fos-Fos, Fos-Jun, and Jun-Jun interactions. However, the fluorescence signal from Fos and Jun heterodimerization was notably weaker compared to their respective homodimerization signals (Fig 4C–4H). To confirm the specificity of these interactions, we expressed individual BiFC-tagged monomers in these neurons and did not detect any BiFC signal, validating that the observed fluorescence resulted from protein-protein interactions (S3A Fig).

To assess whether these interactions change following axotomy, we severed axons and monitored the BiFC signal at different time points. The nuclear BiFC signal between Fos-CC and Jun-VN progressively diminished after injury (Fig 4C and 4D) consistent with functional data indicating they do not work together in the axon injury response. Jun is known to homodimerize in mammalian cells [51,52], although this is not well-established in Drosophila, so we also examined Jun-Jun interactions during the axon injury response. Strong nuclear BiFC between Jun-CC and Jun-VN was seen in unin-jured neurons and at 6 hours after injury, and this was reduced at 24 hpi and even more so at 48 hpi (Fig 4E and 4F). To

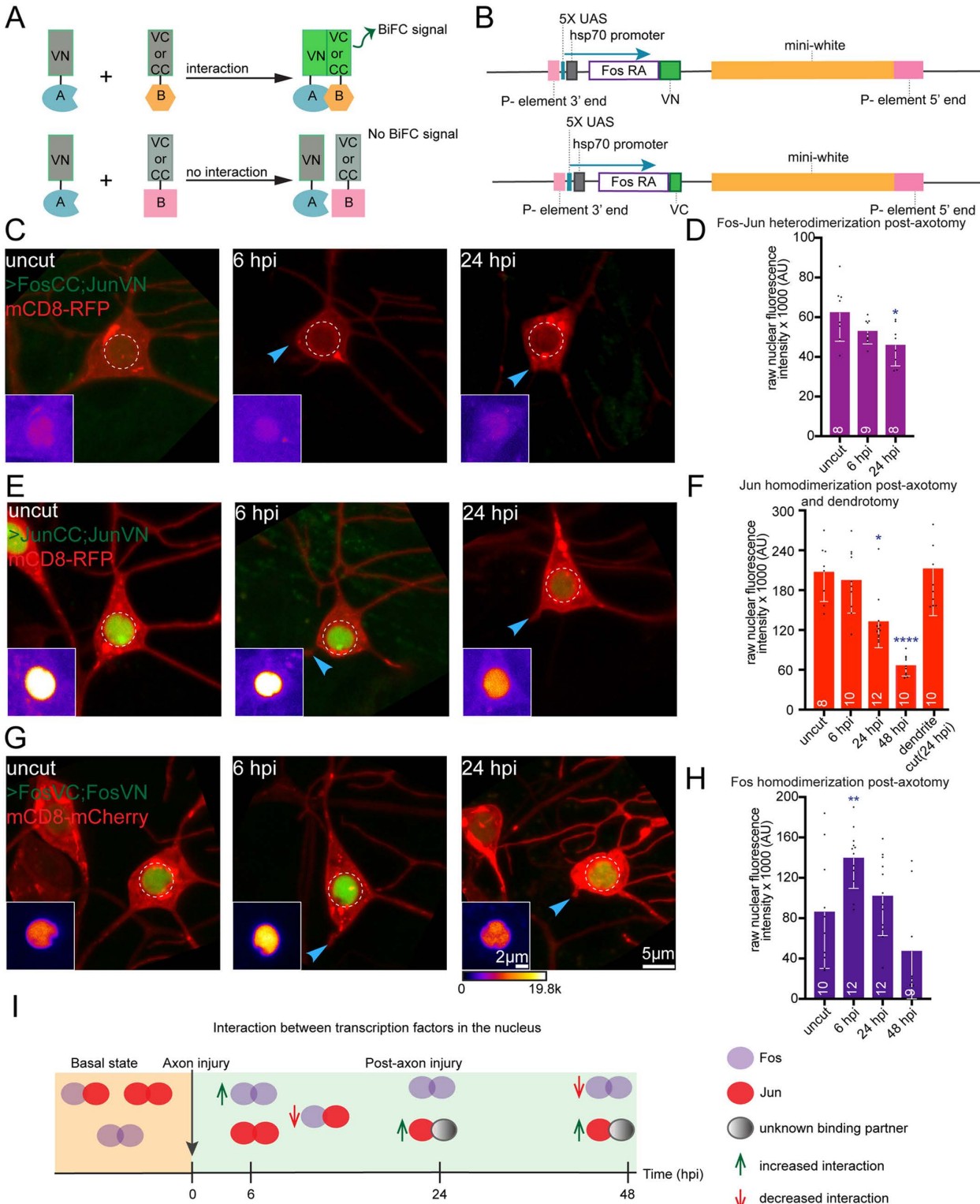

**Fig 4. Fos does not interact with Jun but homodimerizes post axotomy.** (A) Schematic illustrating the BiFC assay where protein tagged with the N-terminus of Venus (VN) and protein tagged with either C-terminus of Venus (VC) or Cerulean (CC) interact, allowing visualization of this interaction via fluorescent protein reconstitution (top). If proteins attached to the BiFC fragments do not bind one another, then no fluorophore is generated (bottom).

(B) Diagram of the BiFC constructs (Fos-VN and Fos-VC) generated to investigate Fos homodimerization. (C) Representative images of FosCC-JunVN BiFC in Class I sensory neurons at the basal state, 6 and 24 hpi. (D) Quantification of raw nuclear fluorescence intensity from FosCC-JunVN BiFC, where Fos-Jun heterodimerization is weak at basal state and this diminishes significantly by 24 hpi. (E) Example images of JunCC-JunVN BiFC at basal state, 6 and 24 hpi. (F) Quantification of raw nuclear fluorescence intensity of JunCC-JunVN, showing Jun homodimerization reduction by 24 hpi and further decrease by 48hpi. Similar reduction in homodimerization signal is not observed at 24 hpi post dendrite cut. (G) Images of FosVC-FosVN at basal state, 6 and 24 hpi. (H) Quantification of raw nuclear fluorescence intensity of FosVC-FosVN, showing increase in homodimerization of Fos at 6 hpi and gradual decline at 24-48 hpi. Blue arrows in (C), (E), (G) show axotomy site. Insets show the region of interest (dashed circle) visualized using fire LUT. (I) Summary model of Fos and Jun interaction dynamics before and after axon injury. At the basal state, Fos and Jun form homodimers. After injury, this homodimerization increases for Fos by 6 hpi, while reduces for Jun by 24 hpi. This reduction in Jun homodimerization is likely due to interaction with an unknown binding partner involved in the axon injury response. Fos and Jun heterodimerization is relatively weak at the basal state which reduces further after axon injury, suggesting that Fos and Jun do not interact during injury responses. Statistical analyses in (D), (F) and (H) were performed using Kruskal-Wallis one way ANOVA test. Error bars represent SD and numbers in bars are number of animals tested. *, $p < 0.05$, **$p < 0.01$, ****$p < 0.0001$.

determine whether this reduction was specifically due to axotomy rather than potential photobleaching effects from laser injury, we performed a dendrite severing control in these neurons and examined the BiFC signal at 24 hpi. In this case, the BiFC signal remained unchanged (Fig 4F). Thus, axon injury signaling seems to specifically cause Jun homodimers to dissociate during the later phase of the injury response and Jun likely has a different bZip partner at this time (Fig 4I).

Whether Fos forms functional homodimers has not been established in mammals or flies, but because Fos was the only transcription factor we identified with a role in the initial axon injury response, we tested interactions between Fos-VN and Fos-VC. At the basal state, we detected a Fos homodimerization signal. This signal increased substantially at 6 hpi and then declined at later time points (Fig 4G and 4H) in a temporal pattern that closely mirrors the nuclear dynamics of endogenously tagged Fos (Fig 3C). From this, we conclude that Fos is capable of homodimerization in Drosophila sensory neurons, and this homodimerization is favored during the early axon injury response (Fig 4I). Taken together, the functional and fluorescence data suggests that Fos homodimerizes in response to axon injury and orchestrates the neuroprotective phase. Subsequently, Jun homodimerization is reduced, but it does not partner with Fos.

### DLK/JNK activation disrupts Jun homodimerization independently from Fos

We considered two hypotheses that could account for dissociation of Jun homodimers after axon injury. Either phosphorylation of Jun could disrupt homodimerization, or another binding partner could be upregulated, perhaps by early Fos activity, and compete for interaction. We considered the second possibility less likely as Jun is overexpressed in the BiFC assay and the competing binding partner would be at endogenous levels, and because Jun is known to be phosphorylated by JNK [53,54]. To determine whether the reduction in homodimerization is dependent on DLK signaling, we knocked down DLK using RNA interference (RNAi) and assessed Jun BiFC at 24 hpi. In the absence of DLK, Jun-Jun BiFC signal remained high after axon injury (Fig 5A and 5B). Similarly, a dominant-negative JNK blocked the injury-induced reduction in BiFC signal (Fig 5B). In contrast, Jun-Jun BiFC after injury was even lower than in control cells when Fos was reduced by RNAi (Fig 5A and 5B), perhaps because Fos activity normally tempers JNK signaling through transcriptional activation of puc. Indeed, we know that this Fos RNAi line blocks puc expression (Fig 2A and 2B). These results suggest that the reduction in Jun homodimerization post-injury is dependent on DLK/JNK signaling, but not Fos-mediated transcription.

To test more directly whether Jun phosphorylation can disrupt homodimerization, we used a competition assay. We hypothesized that overexpressing untagged Jun would compete with BiFC fragment-tagged Jun, thereby reducing the BiFC signal under basal conditions. If phosphorylation destabilizes Jun homodimers, a phosphomimetic Jun should not compete as effectively as untagged wildtype Jun. As predicted, overexpression of wildtype Jun reduced the BiFC signal at the basal state (S3B and S3C Fig). Overexpression of a phosphomimetic Jun variant [55] in which six serine or threonine residues were substituted with aspartic acid was less effective at reducing the BiFC signal (S3B and S3C Fig). This result supports the idea that phosphorylation of Jun disrupts Jun homodimerization, although with the caveat that

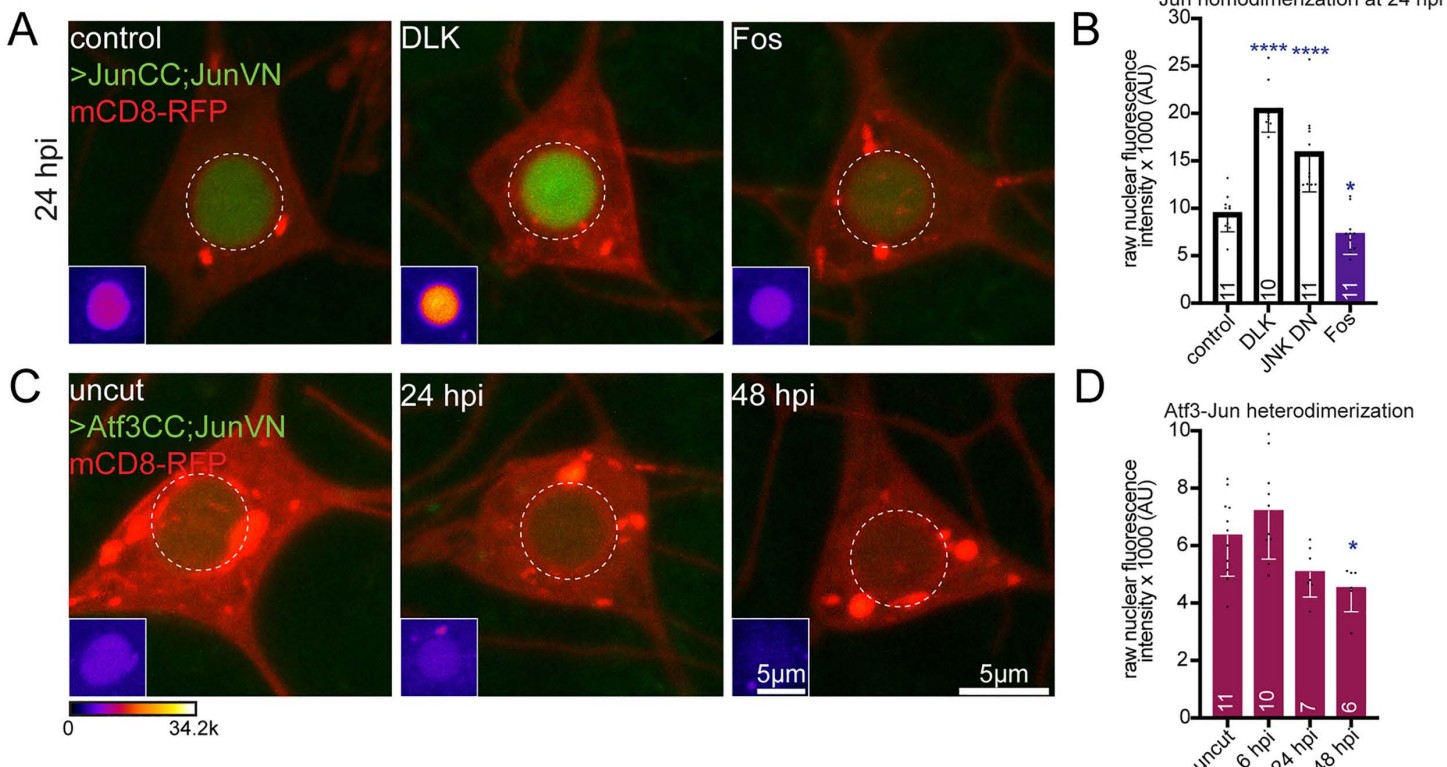

**Fig 5. DLK/JNK signaling pathway promotes Jun homodimer dissociation.** (A) Example images showing the effect of DLK and Fos knock down on Jun homodimerization at 24 hpi in ddaE neurons using BiFC assay. (B) Quantification of raw nuclear BiFC fluorescence intensity in DLK and Fos RNAi and JNK DN background at 24 hpi. Jun homodimerization reduction at 24 hpi is dependent on DLK and JNK but not Fos. (C) Representative images showing Atf3CC-JunVN heterodimerization in ddaE neurons at basal state and different time points post axotomy using BiFC assay. (D) Quantification of nuclear BiFC fluorescence intensity of Atf3CC-JunVN showing weak interaction at the basal state which reduces further by 48 hpi. Insets in (A, C) show region of interest (indicated by dashed white circle) rendered with fire LUT for the BiFC channel. Statistical analyses in (B) has been performed using Mann-Whitney test while that in (D) has been performed with Kruskal-Wallis one way ANOVA test. Error bars represent SD and numbers on the bars are number of animals tested. *, $p < 0.05$, ****$p < 0.0001$.

we do not know whether wildtype and phosphomimetic Jun are expressed at identical levels even though they are both UAS-controlled.

Since Jun-Jun and Jun-Fos interactions were reduced after axon injury, another bZip transcription factor must work with Jun during axon regeneration. Based on the previous finding that neurite growth is enhanced in cells expressing Atf3 and Jun together [27], we hypothesized that Jun might heterodimerize with Atf3 after axon injury. We therefore co-expressed Jun-VC and Atf3-CC and measured the BiFC signal at different time points post-axotomy. We observed low but detectable dimerization at the basal state; however, the BiFC signal decreased by 24 hpi and was further reduced by 48 hpi (Fig 5C and 5D). This suggests that Jun does not stably interact with Atf3 following axon injury. To further test whether Jun and Atf3 function together in the axon injury response, we co-overexpressed both transcription factors and assessed axon regeneration at 96 hpi. There was no enhancement of growth compared to the control (Fig S1D and S1E), consistent with lack of interaction between Jun and Atf3 in this context.

Given that Jun does not seem to bind Fos, Jun, or Atf3 post-injury in our system, we conducted a broader genetic screen to identify potential Jun binding partners in the axon injury context. We selected eleven candidate TFs based on Jun binding in a Drosophila transcription factor interactome study [56]. We also included five Fos binding proteins, as Jun

interactors often overlap with Fos interactors. To assess whether any of these candidates play a role in the axon injury response, we knocked them down and examined growth at 96 hpi. Out of the sixteen candidates, only two—Gt (giant) and Optix—showed a significant reduction in axon regeneration (S4A Fig). Using BiFC, we tested whether either of these interacted with Jun post axotomy. In both cases, we observed a reduction in BiFC signal for heterodimerization with Jun at 24 hpi (S4B–S4E Fig). Thus, although Gt and Optix are required for axon regeneration, we could not find any evidence that they are the missing binding partner for Jun. In summary, our data suggests that Jun phosphorylation after axon injury may disrupt its homodimerization, likely allowing it to partner with another bZip transcription factor. However, we have not been able to identify this partner.

### STAT nuclear accumulation post-axon injury is negatively regulated by Fos and requires Jun

Having established that Fos and Jun regulate axon injury responses at distinct time points, we next investigated the temporal dynamics of STAT after axon injury. To do this, we used GFP-tagged STAT expressed from its own regulatory region (https://flybase.org/reports/FBal0268865) and examined its localization pattern post-injury. Under basal conditions and 6 hpi, STAT-GFP fluorescence was higher in the cytoplasm than the nucleus (Fig 6A). However, by 24 hours post-injury (hpi), the nuclear-to-cytoplasmic ratio (N/C ratio) increased (Fig 6A and 6B). This timing of nuclear entry is consistent with STAT not playing a role in the early transcriptional response reported on by puc (Fig 2A and 2B).

To determine whether nuclear entry of STAT could be used to position it relative to other injury signaling proteins, we assessed whether it depended on DLK/JNK. Indeed, DLK knockdown or expression of dominant-negative JNK prevented the nuclear accumulation of STAT-GFP (Fig 6C and 6D). STAT is almost always linked to the non-receptor tyrosine kinase JAK in signal transduction pathways [57–59]. Increased JAK/STAT signaling has been associated with improved regeneration of retinal ganglion cell (RGC) axons [60] and knockout of Socs3, a JAK/STAT signaling inhibitor, increases RGC axon regeneration [61]. However, loss of function experiments to test for a direct role of JAK in axon injury signaling have not been performed. In Drosophila there is one JAK (hop) and one STAT (Stat92E) [62–65] making this an ideal system to test whether JAK is required upstream of STAT after axon injury. RNAi knockdown of JAK did not block the increase in N/C ratio of STAT-GFP at 24 hpi (Fig 6C and 6D). To confirm this result, we monitored STAT-GFP in *JAK* mutant animals and similarly found that it did not affect the nuclear accumulation of STAT-GFP at 24 hpi (Fig 6C and 6D). To test whether JAK might play a role in the axon injury response separate from nuclear targeting of STAT, we measured axon regeneration in *JAK* mutant animals. There was no reduction in growth in this background (S5A and S5B Fig). These results suggest that DLK/JNK signaling drives activation of STAT in response to axon injury, rather than receptor signaling through JAK. To further test whether DLK/JNK signaling drives the STAT response to axon injury without JAK, we overexpressed constitutively active Hep [66], a MAPKK that lies downstream to DLK and phosphorylates and activates JNK [67], and assessed the N/C ratio of STAT-GFP at the basal state. A significant increase in nuclear STAT-GFP was observed compared to the control (Fig 6I and 6J). Thus, DLK/JNK signaling alone is sufficient to promote STAT-GFP nuclear accumulation, however we do not know whether it acts directly on STAT or through a downstream transcription factor like Fos or Jun.

To test whether STAT nuclear entry is regulated by Fos or Jun, we reduced their levels and assayed STAT-GFP localization. Knocking down Fos seemed to partially block STAT-GFP localization post injury (Fig 6D). This could be due to the incomplete depletion of Fos by RNAi. To address this, we combined Fos RNAi and one copy of a *Fos* mutant to further reduce Fos levels. Surprisingly, strong nuclear STAT-GFP signal was observed in uninjured neurons in this background (Fig 6E and 6F) suggesting that Fos negatively regulates STAT nuclear entry. No further increase was induced by axon injury (Fig 6E and 6F). To further probe the relationship between Fos and STAT, we overexpressed Fos and tracked the N/C ratio of STAT-GFP. At baseline, the STAT N/C ratio was not changed by Fos overexpression (Fig 6J), but it blocked the increase after axon injury (Fig 6G and 6H). Jun overexpression had no effect, suggesting specificity for Fos. These results suggest that Fos inhibits nuclear entry of STAT and that Fos downregulation could facilitate STAT nuclear entry during the transition from early to late injury responses. In contrast to Fos, Jun reduction by RNAi or RNAi/mutant

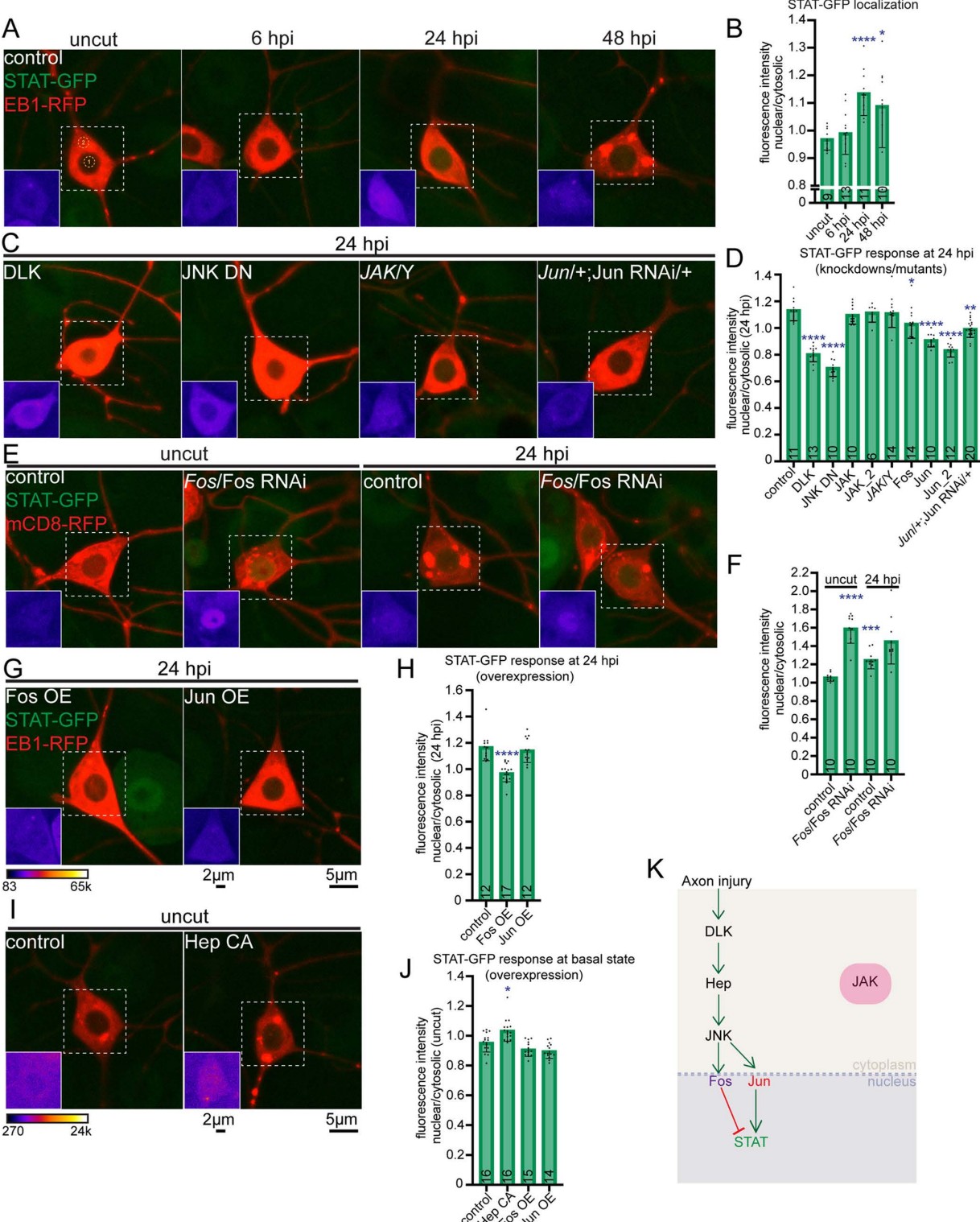

**Fig 6. Axon injury-induced STAT nuclear accumulation requires DLK/JNK/Jun signaling.** (A) Representative images of endogenously tagged STAT-GFP in ddaE neurons pre- and post- axon injury (6, 24, 48 hpi). The numbers 1 and 2 labeled in yellow circles indicate region of interest used for nuclear and cytoplasmic fluorescence intensity measurements, respectively. (B) Quantification of nuclear to cytoplasmic STAT-GFP fluorescence

intensity at the indiacted time points shows increased nuclear localization at 24 hpi. (C) Representative images of STAT-GFP localization at 24 hpi in different genetic conditions, including RNAi-mediated knockdown, mutant, dominant negative forms and combination of mutant allele and RNAi. (D) Quantification of nuclear to cytoplasm STAT-GFP fluorescence intensity at 24 hpi across genetic conditions including two different JAK RNAi knockdown transgenes (JAK, BDSC 31919 and JAK_2, BDSC 31699) and male larvae in which the only copy of JAK (hop) is a null mutant allele (*hop [2]*/Y), two different Jun RNAi transgenes (Jun, BDSC 31595 and Jun_2, VDRC 107997, and the Jun RNAi in the background of a heterozygous *Jun* null mutant allele (*Jra[IA109]*). The Jun mutant chromosome contains additional background mutations that are listed in S1 Table. (E) Representative images of STAT-GFP acquired pre-injury and at 24 hpi in Fos loss-of-function background (combination of *Fos* mutant allele and Fos RNAi). (F) Quantification showing that reduced levels of Fos enable nuclear entry of STAT. (G) Representative images showing STAT-GFP localization at 24 hpi in backgrounds overexpressing Fos or Jun Class I sensory neurons. (H) Quantification of nuclear to cytoplasm fluorescence intensity of STAT-GFP at 24 hpi in the overexpression conditions. (I) Example images of STAT-GFP localization at the basal condition in the presence of constitutively active Hep. (J) Quantification showing increased nuclear localization of STAT-GFP at the basal state in the Hep constitutively active (CA) background compared to the control. In contrast, Fos and Jun overexpression do not show effects. Insets in (A, C, E, G, I) show region of interest, indicated using a white square, visualized using fire LUT for the STAT channel. (K) Schematic model summarizing the pathway: axon injury triggers DLK/Hep/JNK which can bifurcate into two signaling axes- one through Fos and the other through Jun. The Fos signaling axis inhibits STAT nuclear translocation while the Jun axis promotes STAT nuclear translocation at 24 hpi. Thus, Fos downregulation is required to activate DLK/Hep/JNK/Jun signaling arm for STAT nuclear localization. This pathway functions independent of JAK signaling. Kruskal-Wallis one way ANOVA test was used for statistical analyses. Error bars represent SD and numbers on bars are number of animals tested. *$p < 0.05$, **$p < 0.01$, ***$p < 0.001$, ****$p < 0.0001$.

combination blocked nuclear entry of STAT-GFP after axon injury similar to reduction of DLK or JNK (Fig 6C and 6D) suggesting that DLK/JNK do not act directly on STAT after injury but require Jun-mediated transcription. Taken together, this data places DLK/JNK/Jun in a positive regulatory pathway upstream of STAT, while Fos negatively regulates it and JAK plays no role in this context (Fig 6K).

## Atf3 nuclear dynamics is regulated by DLK, Fos, Jun and STAT

Having mapped the relationship between three of the four DLK-linked transcription factors we chose to study, we next wished to include Atf3. One potential role for Atf3 was as the Jun-binding partner, but the BiFC results (Fig 5C and 5D) did not support this. Knocking down Atf3 had no detectable effect on puc-GFP (Fig 2A and 2B) or STAT-GFP (S5C and S5D Fig) nuclear levels and overexpressing Atf3 had no effect on axon regeneration (S6A and S6B) suggesting Atf3 acts at a later stage of the injury response. To investigate this possibility, we examined the nuclear localization dynamics using Atf3-GFP expressed from its own regulatory sequences (https://flybase.org/reports/FBti0150285.html).

At baseline, Atf3-GFP was readily detected in the nucleus (Fig 7A). However, its fluorescence intensity decreased by 6 hpi in the cut ddaE cell compared to the adjacent uninjured ddaD cell and further reduced by 24 hpi. Interestingly, by 48 hpi, nuclear fluorescence intensity in the ddaE cell increased again, suggesting a biphasic nuclear response (Fig 7A and 7B). Although we do not understand how this localization pattern maps on to Atf3 function, the injury-induced change allowed us to determine what players in the DLK signaling cascade it responds to.

To place Atf3 nuclear localization within the DLK signaling cascade, we generated a tester line containing a red cell shape marker, Atf3-GFP and dicer2 and crossed it with RNAi lines (validated in S6C and S6D Fig). In DLK RNAi neurons, nuclear Atf3-GFP did not decrease at 24 hpi, indicating that DLK signaling is required for its nuclear exit. We then tested whether this regulation occurs downstream of other transcription factors involved in axon injury responses. Knocking down Fos, Jun, and STAT similarly blocked the 24 hpi decrease in nuclear Atf3-GFP, suggesting that Atf3 is regulated downstream of these factors (Fig 7C and 7D).

To further explore whether Atf3 nuclear localization responds to other injury-regulated transcription factors, we overexpressed Fos and Jun and measured nuclear Atf3-GFP levels at basal state. We observed a reduction in nuclear Atf3-GFP upon Fos or Jun overexpression, supporting the idea that both transcription factors act as negative regulators of nuclear Atf3 (Fig 7E and 7F).

Taken together, these results support a model in which Atf3 acts as a late-phase regulator of axon regeneration, with its nuclear dynamics controlled by an upstream transcriptional network involving Fos, Jun, STAT, and DLK signaling (Fig 7G).

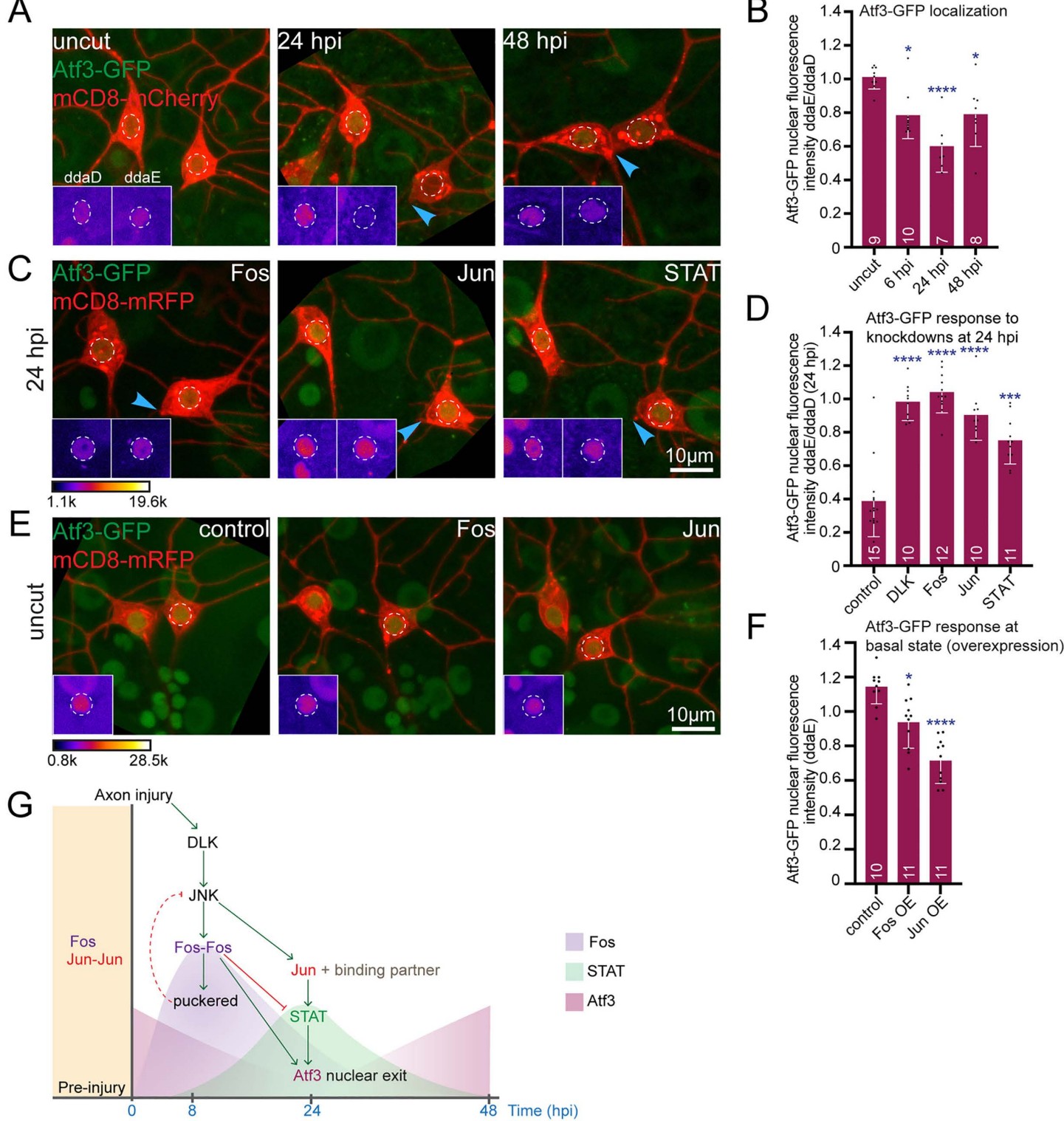

**Fig 7. Axon injury induces DLK-dependent biphasic changes in nuclear localization of Atf3.** (A) Representative images showing endogenously tagged Atf3-GFP in Class I sensory neurons at the basal state and 24 and 48 hpi. (B) Quantification of nuclear fluorescence intensity of Atf3-GFP in ddaE normalized with that of adjacent ddaD before and after injury (6, 24, 48 hpi). Atf3-GFP nuclear localization reduces by 24 hpi and again increases

by 48 hpi. (C) Example images showing effect of knock down of DLK, Fos, Jun and STAT on Atf3-GFP localization at 24 hpi. (D) Quantification of nuclear fluorescence intensity of Atf3-GFP in ddaE (injured) neurons normalized to ddaD (uninjured) showing that knockdown of DLK, Fos, Jun, STAT blocks nuclear exit of Atf3 at 24 hpi. (E) Images showing effect of overexpressing Fos and Jun on nuclear Atf3-GFP levels at basal state. (F) Quantification of raw nuclear fluorescence intensity of Atf3-GFP in ddaE showing that Fos and Jun overexpression reduces nuclear Atf3-GFP level at basal conditions. Insets in (A,C and E) indicate the region of interest, marked with white dotted circle, visualized using fire LUT for the Atf3 channel. (F) Schematic model summarizing DLK-dependent temporal cascade of transcription factors induced by axon injury. DLK promotes nuclear accumulation and homodimerization of Fos which regulates the early injury response. As the response progresses, DLK/JNK signaling triggers switching of Jun homodimer to a heterodimer state, with an unknown binding partner, to regulate later injury responses. Concurrently, nuclear Fos levels decline by 24 hpi to allow DLK/JNK/Jun-dependent entry of STAT during this time. DLK also drives nuclear exit of Atf3 at 24 hpi, contributing to the dynamic regulation of injury-induced transcriptional program. The red dotted line shows a relationship that has not been shown in this study but is supported by prior studies. The statistical analyses were performed using Kruskal-Wallis one way ANOVA test and error bars represent SD; numbers on bars are number of animals in each condition. *$p < 0.05$, ***$p < 0.001$, ****$p < 0.0001$.

## Discussion

Using Drosophila as a model system we show that key transcription factors activated by DLK in response to axon injury are organized into two modules. The early module is mediated by Fos, which homodimerizes in response to axon injury signaling. The later module is controlled by Jun, which responds to DLK signaling independently from Fos. Jun initiates a linear cascade that controls nuclear localization of STAT and Atf3, the other two transcription factors functionally linked to axon regeneration in mammals. While Jun can respond to injury signaling without input from Fos, the two downstream transcription factors are regulated by Fos (Fig 7G). The first module corresponds to a neuroprotective phase characterized by high microtubule dynamics and resistance to degeneration [29]. The second module corresponds to the outgrowth step of axon regeneration. We previously showed that prolonged activation of the neuroprotective step, for example by Fos overexpression, inhibits axon regeneration [31]. This inhibition could be due to the negative effect of Fos on STAT nuclear entry (Fig 6E–6H). Intriguingly, although neuroprotection inhibits regeneration, Fos reduction does not itself enhance regeneration (Fig 1C). The positive role of Fos in axon regeneration could be mediated by its requirement to regulate nuclear localization of Atf3 (Fig 7C–7F). These findings contribute to a deeper understanding of how DLK orchestrates complex transcriptional networks to control axon injury responses.

The linear dependency of the axon outgrowth transcriptional module (Jun>STAT>Atf3) is somewhat surprising. In a previous study that examined the relationship between STAT and DLK, axonal STAT was phosphorylated in DLK knockout mice, but its increase in the somatic region was reduced [14]. The phosphorylation was assumed to be mediated by JAK, and the role of DLK was proposed to be through regulation of STAT transport to the cell body rather than indirectly through Jun [14]. Our results indicating that JAK is not required for axon regeneration and that Jun acts upstream of STAT suggest that deeper investigation of STAT activation in other systems will be important. In mammals there are multiple JAK kinases, and loss of function studies to demonstrate their role in axon regeneration have not been performed to our knowledge. The temporal relationship between Jun and STAT is supported by transcriptome analysis of DRG neurons at early timepoints after axon injury; c-Jun transcripts are elevated before those for STAT3 [68]. How the Atf3 nuclear exit then reentry that we observe after axon injury relates to function and overall levels is unclear at this point, but we made use of its injury-dependent nuclear exit as an assay to determine whether it responds to the other transcription factors. There is data from mammalian systems to show that Jun can increase Atf3 expression [27] supporting the linear relationship between these two transcription factors suggested by our study, in contrast there is not previous evidence suggesting STAT might function upstream of Atf3. While this effect may be indirect, our data does demonstrate that without STAT, Atf3 does not respond normally to injury. One piece of evidence from mammalian studies that could support this linear relationship with Atf3 as the output of the other transcription factors is a study demonstrating that expression of Jun and STAT with Atf3 is no more effective at promoting regeneration than Atf3 alone [69]. Thus, although the linear relationship between these transcription factors is not expected, there is data from mammalian systems to suggest it is worth investigating in other contexts.

While the second transcriptional module that is associated with axon regrowth links three of the core transcription factors known to be functionally important in mammals, the first module relies on Fos to control neuroprotection. Is Fos also required for axon injury responses in other systems? Within Drosophila, Fos coordinates injury responses, including a neuroprotective program, in motor neurons [15,70] as well as the sensory neurons used here. In mammals, based on transcript clustering, different phases of the axon injury response have been defined for DRG neurons, with a major switch occurring around one day after injury [71]. High levels of Fos transcripts were observed 8-28h after DRG axon injury [68] positioning Fos within the early stage. A functional readout for this early stage has not been developed, so what Fos might be doing is not known. It is intriguing to speculate that mammalian neurons may also be stabilized at this time. One of the hallmarks of neuroprotection after injury in Drosophila is increased microtubule dynamics. A strong increase in microtubule dynamics has been detected in intercostal nerves 1–2 days after axon injury [72] hinting that early injury responses could be similar across species.

The mechanisms underlying the temporal regulation of transcription factor activation after axon injury remains poorly understood. Fos and Jun are both JNK substrates, and phosphorylation of Jun seems to drive homodimer dissociation, but this occurs after Fos homodimerizes and upregulates puc. Why is Jun not activated at the same time as Fos? It has been proposed that different JNK outputs are influenced by length and strength of JNK activation [73], and that different scaffolding proteins can coordinate distinct outputs from DLK [74], but it is unclear how the delay in Jun dissociation works here. It is easier to conceptualize how the second module could respond to the first one below Jun. Fos can negatively regulate its own expression [75] and puc MAP kinase phosphatase is induced to reduce JNK signaling. Overexpressing Fos prevents the early neuroprotective phase from being turned off and reduces axon regeneration [31]; this could be because high levels of Fos prevent STAT from entering the nucleus (Fig 6).

One intriguing aspect of regulation we uncovered was the change in binding partners in response to axon injury signaling. Most large-scale studies to map binding interactions between transcription factors do so in only one cell state [56], and our results suggest that in different cellular contexts these interactions could be quite distinct. Regulation of binding partners by phosphorylation is particularly surprising for Jun and Fos. One model for changing composition of AP-1 transcription factors is that before the stimulus Jun-Jun homodimers pre-exist. Their ability to activate transcription is stimulated by JNK. Fos is then transcriptionally upregulated, and it then competes with Jun to form more stable Fos-Jun heterodimers [53]. Phosphorylation is proposed to regulate DNA binding or transactivation, but not bZip partner preference [53,54], so the DLK/JNK-induced homodimerization of Fos and dissociation of Jun was a surprise.

Overall, our findings provide foundation for future studies aimed at understanding the intricate network of transcriptional regulation in axon injury responses. Expanding these studies to include downstream gene regulation by these transcription factors could reveal novel targets for promoting neuroprotection and axon regeneration in injured neurons.

## Materials and methods

### Drosophila stocks and maintenance

Fly lines were grown in bottles and vials with standard fly media containing yeast, cornmeal, sucrose, dextrose, and agar, and maintained at 25°C. All fly lines used in this study were obtained from Bloomington Drosophila Stock Center, Vienna Drosophila Resource Center or FlyORF. The specifics of all the lines used in this study can be found in S1 Table.

Virgin females from tester lines (except for transgenes on X chromosome) were crossed with males from RNAi, dominant negative, mutant, transcription factor-reporter or overexpression lines. The experimental crosses were maintained at 25°C for three days before imaging third instar larvae. The tester lines used are as follows:

Fig 1B–1D: 221-Gal4, UAS-EB1-GFP, UAS-dicer2-nls$^4$BFP/Tm6

Fig 2A: UAS-dicer2, UAS-mCD8-RFP/CyO; 221-gal4, puc-GFP/TM6

Fig 2C: 221-Gal4, UAS-EB1-DFP

Fig 5A and 5B: TMC-Gal4-UAS-JunCC-UAS-dicer2-CytoBFP/CyO; UAS-bJunVN/Tm6

Fig 6E and 6F: TMC-Gal4, UAS-mCD8-RFP/CyO; Kay RNAi/Tm6 and TMC-Gal4, UAS-mCD8-RFP/CyO; γ-tubulin37C RNAi/Tm6

S2B and S2C Fig: TMC-Gal4, UAS-mCD8-RFP, UAS-dicer2-CytoBFP/CyO; Fos-mNG.

**Generation of Fos and Jun transgenic lines**

A. CRISPR-mediated knock-in of Fos and Jun tagged with mNeonGreen (mNG)

The Drosophila first generation CRISPR repair template 1GCT-pHD-dsRed was used to make 1GCT-1XmNG-dsRed as previously described [76,77]. The 1GCT-1XmNG-dsRed was subsequently digested with restriction enzymes SrfI and SphI.
The Jun coding region was cloned by In-Fusion as a 1500 bp PCR fragment amplified from Drosophila genomic DNA using the following primers:

JUN1F1: GAAAGACTGGGCCTTTCGCCCGGGCGATCTATTGCATCTTCTTTTACTGCCAAACATCAAG

JUN1R1: CACGCGTGGTACCGCCGGCATGCTGTTGGTCTGTCGAGTTCGGCGGCACCGTGCAGCCCG

The resulting construct 1GCT-JUN_S1_1XmNG-dsRed was digested with SrfI, and 1200 bp of Jun upstream sequences were cloned into the vector by In-Fusion cloning. This was done as a PCR fragment amplified using the following primers:

JUN2F1: GAAAGACTGGGCCTTTCTCGCGAGGCGGCAGCAGCCAACTAGAGTTTACCAG

JUN2R1: AGAAGATGCAATAGATCGCGTCGGTAAATAGGAAAGGAAATATATTTTTCGG

The product 1GCT-JUN_S2_1XmNG-dsRed was digested with AsiSI, and 230 bp of the Jun terminator sequence was cloned as a PCR fragment amplified using the following primers:

JUN3F1: CTGTACAAGAGCTCATATAACATTTGGAGTTGTCAGCCGGGGAG

JUN3R1: GCCGCGATGTCGACTCTTCCTCCTCCAGGAGTTCGCTGTTAGTTAGG

The resulting construct 1GCT-JUN_S3_1XmNG-dsRed was digested with AscI, and 1050 bp of downstream flanking sequence was cloned as a PCR fragment amplified using the following primers:

JUN4F1: AAGATCTCCATGCATAATGCTCTACATCCTCGCGGACGACGCAACCTG

JUN4R1: GAGCTGCAGAAGGCCTAGACCGCAAGGGTCTGGCTTGGAAACAGCGCATTG

The final construct 1GCT-JUN_S4_1XmNG-dsRed was sent to BestGene as repair template for CRISPR along with the guide RNA construct that was generated by In-Fusion cloning of a 250 bp PCR product into pCFD5 (Addgene plasmid 73914) digested with BbsI.

The PCR fragment was amplified using pCFD5 as template using the following primers:

JUNgF1: GCGGCCCGGGTTCGATTCCCGGCCGATGCATCTTTATCACTCCTGGGCGTGTTTTA-GAGCTAGAAATAGCAAG

JUNgR1: ATTTTAACTTGCTATTTCTAGCTCTAAAACGAGGCTGTTCGTTATCGCTCTGCACCAGCCGGGAATC-GAACCC

The identical methodology was used for making the Fos CRISPR repair template with the following primers being used in the four- step cloning strategy:

Step1:

FOSIF1: TCGAAAGACTGGGCCTTTCGCCCGGGCGCAACACTGCGTATGCGTGATATATGTATACTG

FOS1R1: CACGCGTGGTACCGCCGGCATGCTGTAAGCTGACCAGCTTGGACGGCTCGGCG

Step2:

FOS2F1: GAAAGACTGGGCCTTTCATGAAAAATCTCAACGGTAGGACGCACAATGCGTG

FOS2R1: CGCATACGCAGTGTTGCGGGAATTTATTCTTATCACATGAGTAATTTG

Step3:

FOS3F1: CTGTACAAGAGCTCATATAACCGGACCGGATCTGGGACGGGGATG

FOS3R1: GCCGCGATGTCGACTCTTATCAATCGGTTTGGTTTTGTATAGATTAAAATTTTG

Step4:

FOS4F1: AAGATCTCCATGCATAAAAGACGATTTGTGGAATTATTTCTGAAG

FOS4R1: GAGCTGCAGAAGGCCTACAATGTTGCTGTCCAGGACGAAGGAGCCTG

The guide RNA construct for Fos was also constructed in the identical manner as for Jun but utilizing PCR product amplified with the following primers:

- FOSgF1: GCGGCCCGGGTTCGATTCCCGGCCGATGCAAAACATCCTTAAGCTATTCGGTTTTA-GAGCTAGAAATAGCAAG

- FOSgR1:ATTTTAACTTGCTATTTCTAGCTCTAAAACTTATAAGTGCATTTATTTGCTGCACCAGCCGGGAATCGAACCC

B. Fos split-YFP construct for BiFC assay

This was generated from previously constructed UAS- γ-tubulin-split-YFP constructs which were made as follows:
The following primers were used to PCR amplify the N-terminus of YFP off of pEYFPC1GgVcl (Addgene plasmid 46279)

N-termYFPF1:

TACCACGCGTGGCCGGCCTGCTAGCATGGTGAGCAAGGGCGAGGAGCTGTTCACCG

N-termYFPR1: GTTCCTTCACAAAGATCCTCTAGATTAGTCGGCCATGATATAGACGTTGTGGCTGTTGTAGTTG

The PCR product was used to replace RFP in the previously made construct UAS-ROR-RFP [78]. This was done by digesting UAS-ROR-RFP with NheI-XbaI and cloning the 468 bp PCR product by In-Fusion.

The following primers were used to PCR amplify the C-terminus of Venus off Venus-iLID-CAAX (Addgene plasmid 60411):

C-termYFPF1: CCACGCGTGGCCGGCCTGCTAGCGCCGACAAGCAGAAGAACGGCATCAAG

C-termYFPR1: GTTCCTTCACAAAGATCCTCTAGATTACTTGTACAGCTCGTCCATGCCGAG

C-terminal YFP was used to replace RFP in UAS-ROR-RFP. This was done by digesting UAS-ROR-RFP with BmtI-XbaI and cloning the C-term YFP PCR product by In-Fusion. Once the UAS-ROR-split-YFP constructs were made, ROR was replaced with γ-tubulin by first PCR amplifying γ-tubulin using the following primers:

- gTubEcoRIF1: TCTGAATAGGGAATTGGGAATTCAACATGCCAAGTGAAATAATTACTTTGCAGCTTGG
- gTubAcc65IR1: AGCAGGCCGGCCACGCGTGGTACCGGAACCGGCGCTGGTCACAGATCGACTATC

The resulting PCR product was then cloned by In-Fusion into the N- and C- terminal versions of UAS-ROR-split-YFP, in both cases the parental construct was digested with EcoRI and Acc65I.

To generate Fos split-YFP constructs the following primers were used to PCR amplify Fos using cDNA LD16083 as template:

FOSEcoRIF1: GTTTTGTGAATTCTCGCCAACTGGAGAGCAGCAACAATGAAAGTCAAAG

FOSAcc65IR1: AGATCCGGGTACCTAAGCTGACCAGCTTGGACGGCTCGGCGGTGGGCG

The resulting 1800 bp fragment was digested with EcoRI-Acc65I and cloned into EcoRI-Acc65I digested γ-tubulin Split-YFP N- and C-terminal constructs by ligation, replacing γ-tubulin with Fos.

The plasmids were sent to BestGene for injection into fly embryos and the insertions for the split-YFP-construct was mapped to identify the chromosome containing the insertion using balancers. Standard genetic methods were used to generate tester lines using these constructs.

## Axon regeneration assay

Early third-instar larvae were selected and mounted between a slide and a coverslip, secured with Scotch tape. Axons of ddaE neurons expressing EB1-GFP under the control of either 221-Gal4 or TMC-Gal4 in the 5th or 6th hemisegment were identified using a 63x oil objective (NA 1.4) and severed proximally to the cell body using a MicroPoint UV pulsed laser (Andor Technology). Only a single axon was injured per larva. Immediately following injury, the cut cells were imaged using a Zeiss inverted LSM800 confocal microscope.

After imaging, larvae were individually recovered in food caps and incubated at 20°C for 96 hours to slow the larval growth and avoid pupa formation. Larvae exhibiting explosion cuts (bright, auto fluorescent spots at the injury site) were excluded from further analysis. At 96 hours post-injury (hpi), the same cells were reimaged using the Zeiss LSM800 confocal microscope with a 63x oil objective (NA 1.4). Since regenerating axons often extended into the adjacent hemisegment, multiple image stacks were acquired to capture the full axon trajectory.

Maximum intensity Z- projections were generated using Fiji. If multiple images were required to visualize a regenerating axon, they were stitched using the pairwise stitching plugin [79] and processed for quantification in the same software. The following measurements were taken:

1. Regenerating axon length at 96 hpi (R96) and the corresponding length at 0 hpi (R0)

2. Non-regenerating dendritic branch length at 96 hpi (NR96) and the corresponding length at 0 hpi (NR0)

Normalized axon regeneration (AR) (tip growth) was calculated using the following formula:

$$\text{Normalized AR} = R96 - \frac{R0}{(NR0/NR96)}$$

The regenerating axon was confirmed based on flipped microtubule polarity. In cases where the polarity reversal was ambiguous (for example, in Fos RNAi neurons), the dendrite closest to the axon was quantified.

### Live imaging of tagged proteins, injury and image analysis

To assess the temporal dynamics of proteins tagged with fluorescent reporters after axon injury, experimental crosses were maintained at 25°C for three days before injury. Larvae expressing a red cell shape marker (EB1-RFP, mCD8-RFP, or mCD8-mCherry) under the control of 221-Gal4 or TMC-Gal4, along with endogenously tagged proteins and BiFC constructs, were selected for imaging. The axon of ddaE neurons in the 5th or 6th hemisegment was severed proximal to the cell body using a pulsed UV laser, and injured larvae were recovered onto individual food caps. These larvae were incubated at 25°C and imaged at specific time points post injury using a Zeiss LSM800 confocal microscope equipped with a 63x oil objective (NA 1.4):

puc-GFP- 6 hpi

Fos-mNG- 8, 24, 48 hpi

Jun-mNG- 8, 24, 48 hpi

STAT-GFP- 6, 24, 48 hpi

BiFC- 6, 24, 48 hpi

To obtain fluorescence intensity levels at the basal state of the cell, uninjured ddaE neurons in the 5th or 6th hemisegment of three-day-old larvae were also imaged.

Two-channel images were processed using Fiji, and maximum intensity Z-projections were generated. The ellipse tool was used to define the region of interest (ROI), and fluorescence intensity measurements were obtained. For all assays, nuclear fluorescence intensity was measured. Additionally, for the STAT-GFP assay, both nuclear and cytoplasmic fluorescence intensities were quantified.

To normalize fluorescence intensity, the nuclear fluorescence intensity of the injured ddaE neuron was divided by the nuclear fluorescence intensity of the uninjured ddaD neuron in the same hemisegment (ddaE/ddaD ratio) for all assays except STAT-GFP and BiFC assay. For STAT-GFP, fluorescence intensity was quantified as the ratio of nuclear to cytoplasmic fluorescence intensity. For BiFC assay, the raw fluorescence intensity was measured and compared with the controls.

### Statistics and data visualization

All Statistical tests (as mentioned in the Fig legends) were performed using GraphPad Prism. The error bars in the graphs represent standard deviation. The Figs were prepared using Adobe Illustrator.

### Supporting information

**S1 Fig. Co-expression of Jun and Fos impair AR, while co-expression with Atf3 has no effect.** (A) Quantification of effect of multiple STAT RNAi lines on new axon tip growth at 96 hpi. The control dataset has been reused from Fig 1C. (B) Quantification of knockdown efficiency of STAT RNAi lines used in (A) tested using STAT-GFP tester line. (C) Quantification of knockdown efficiency of Atf3 RNAi lines tested using Atf3-GFP tester line. (D) Representative images of proximally axotomized class I ddaE cells at 96 hpi in different co-overexpression genetic backgrounds; all conditions include Jun overexpression. (E) Quantification of the new axon tip growth at 96 hpi, showing that co-expressing Fos and Jun significantly reduce axon regeneration compared to the control where Jun is co-expressed with control (UAS-iBlueberry). To determine statistical significance, Kruskal-Wallis one way ANOVA test was performed. The thick

line represents the median, while the dashed lines represent first and third quartiles in the plot (A,E). Error bars in B and C represent SD. Numbers above the genotype are number of animals tested. *p < 0.05, **p < 0.01, ***p < 0.001, ****p < 0.0001.
(TIF)

**S2 Fig. Fos nuclear accumulation after axon injury is dependent on DLK.** (A) Quantification of Fos-mNG nuclear fluorescence intensity in ddaE normalized to ddaD at basal state, 8 hpi, 24 hpi and 48 hpi using a second Fos-mNG fly line. (B) Representative images showing Fos-mNG at 8 hpi in control and DLK RNAi conditions. (C) Quantification of nuclear Fos-mNG fluorescence in ddaE (injured) normalized to ddaD (uninjured) at 8hpi in control and DLK RNAi. Knockdown of DLK reduces the injury-induced increase in Fos-mNG observed at 8 hpi in control. Kruskal-Wallis one way ANOVA test and Mann-Whitney test were performed in A and C respectively to determine statistical significance. The error bars represent SD. Numbers in the bars represent the number of animals tested. ***p < 0.001, ****p < 0.0001.
(TIF)

**S3 Fig. BiFC controls confirm fragment non-fluorescence without complementation and reveal axon-injury specific reduction in Jun homodimerization mediated by phosphorylation.** (A) Representative images of BiFC fragments attached to Jun or Fos, expressed in Class I sensory neurons showing no fluorescence at the basal state, validating that the fragments do not fluoresce without complementation. (B) Representative images of competition-based BiFC assay assessing the impact of overexpressed wild-type Jun and phosphomimetic Jun on JunVN-JunCC BiFC signal at basal state. (C) Quantification of nuclear fluorescence intensity of Jun homodimer BiFC signal at the basal state shows that phosphomimetic Jun does not compete with JunVN and JunCC as much as wild-type Jun. Insets in (B) show region of interest (indicated by dashed white circle) rendered with fire LUT for the BiFC channel. For statistical analysis, Kruskal-Wallis one way ANOVA test was used. Error bars represent SD; sample number is shown on the bar. *p < 0.05.
(TIF)

**S4 Fig. Gt and Optix are required for axon regeneration but do not interact with Jun post-axon injury.** (A) Genetic screening of RNAi lines demonstrating the impact of their knock down on new axon tip growth at 96 hpi in class I ddaE neurons. Gt and Optix RNAi significantly reduced axon regeneration. The control dataset has been reused from the control in [Fig 1C]. (B) Representative images of GtCC-JunVN BiFC in ddaE neuron, showing reduction in heterodimerization post axon injury. (C) Quantification of nuclear fluorescence intensity of BiFC signal at the basal state and 6, 24, 48 hpi when GtCC was co-expressed with JunVN. (D) Example images of JunCC- OptixVN BiFC in Class I ddaE neurons at the basal state and 24 hpi. (E) Quantification of nuclear fluorescence intensity of BiFC signal, showing that Optix and Jun interaction reduces at 24 hpi. The blue arrows represent cut site, and the white dashed circles represent the region of interest used to quantify BiFC signal in (B) and (D). Statistical analysis in (A) was performed using Kruskal-Wallis one way ANOVA test. In the plot, the thick line represents the median, while the dashed lines represent the interquartile range (A). Error bars represent SD (C, E); sample numbers are indicated for each condition. *p < 0.05, **p < 0.01.
(TIF)

**S5 Fig. JAK is dispensable for axon regeneration and Atf3 is not required for STAT nuclear accumulation after axon injury.** (A) Representative image of axon regeneration (AR) in class I ddaE neuron at 96 hpi in *JAK* mutant after proximal axotomy. (B) Quantification of axon tip growth in *JAK* mutant compared to control. The control dataset has been reused from [Fig 1C]. (C) Example image of STAT-GFP 24 hpi in class I ddaE neuron in Atf3 knockdown background. (D) Quantification of nuclear-to-cytoplasmic STAT-GFP fluorescence intensity showing that Atf3 RNAi does not have any effect on nuclear accumulation of STAT-GFP at 24 hpi. The control used for this comparison has been reused from [Fig]

6D. Blue arrow in (C) show axon cut site, and the inset is visual representation of the regions marked in dashed square in (C), created using fire LUT. In the plot, the thick line represents the median, while the dashed lines represent first and third quartiles (A), error bars represent SD (D), sample numbers are shown for each condition. Kruskal-Wallis one way ANOVA test.
(TIF)

**S6 Fig. Atf3 exits the nucleus at 24 hpi and has no effect on axon regeneration when overexpressed.** (A) Representative image of axon regeneration at 96 hpi when Atf3 is overexpressed in class I sensory neurons. (B) Quantification of new axon tip growth at 96 hpi, showing that Atf3 overexpression has no effect on axon regeneration compared to the control (UAS-iBlueberry) which has been reused from Fig 2D. (C) Atf3-GFP tester validation, showing quantification of Atf3-GFP fluorescence intensity in the nucleus of ddaE normalized to the adjacent ddaD, at basal condition and after ddaE axon was cut at 6, 24 and 48 hpi. The 24 hpi dataset includes a subset of control data in Fig 7D. (D) Representative images showing Atf3-GFP localization at basal state, 6 hpi, 24 hpi and 48 hpi using the Atf3-GFP tester. The blue arrows represent cut site, and the white dashed circles represent the region of interest. Statistical analysis in (A) was performed using Kruskal-Wallis one way ANOVA test and error bars in the graphs represent SD (A). In the plot (C), the thick line represents the median, while the dashed lines represent first and third quartiles. Sample numbers are shown for each condition. $**p < 0.01$, $****p < 0.0001$.
(TIF)

**S1 Table. Reagent list.** This includes details of all the fly stocks used in this study.
(XLSX)

**S2 Table. Datasets for all graphs.** This includes the raw data used in the Figs and supplementary Figs.
(XLSX)

## Acknowledgments

We are grateful to Bloomington Drosophila Stock Center (NIH P40OD018537), Drosophila Vienna Resource Center [80] and FlyORF Zurich ORFeome Project [81] for the stocks used in this study. We are also grateful to Flybase (FB2025_02) which has been a valuable resource for gene expression, function, stocks, and related information [82]. We appreciate the Ratnaparkhi lab, Lai lab, Kango-Singh lab and Uhlirova lab for providing us with fly lines used in this study. We are also grateful to Alex Weiner and Gary Teeters for generating fly stocks and testing them.

## Author contributions

**Conceptualization:** Gibarni Mahata, Li Chen, Melissa M. Rolls.

**Formal analysis:** Gibarni Mahata.

**Funding acquisition:** Melissa M. Rolls.

**Investigation:** Gibarni Mahata, Li Chen, Gregory O. Kothe.

**Resources:** Gregory O. Kothe.

**Supervision:** Melissa M. Rolls.

**Validation:** Gibarni Mahata.

**Visualization:** Gibarni Mahata.

**Writing – original draft:** Gibarni Mahata, Melissa M. Rolls.

**Writing – review & editing:** Gibarni Mahata, Li Chen, Gregory O. Kothe, Melissa M. Rolls.

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
