## [Decision Letter · Decision Letter 0]

2 Jul 2025

PGENETICS-D-25-00607

DLK orchestrates a modular transcriptional response to axon injury with separate roles for Fos and Jun

PLOS Genetics

Dear Dr. Rolls,

Thank you for submitting your manuscript to PLOS Genetics. After careful consideration, we feel that it has merit but does not fully meet PLOS Genetics's publication criteria as it currently stands. Therefore, we invite you to submit a revised version of the manuscript that addresses the points raised during the review process.

Please submit your revised manuscript within 60 days Aug 31 2025 11:59PM. If you will need more time than this to complete your revisions, please reply to this message or contact the journal office at plosgenetics@plos.org. Please include the following items when submitting your revised manuscript:

We look forward to receiving your revised manuscript.

Kind regards,

Hongyan Wang, Ph.D.

Academic Editor

PLOS Genetics

Fengwei Yu

Section Editor

PLOS Genetics

Aimée Dudley

Editor-in-Chief

PLOS Genetics

Anne Goriely

Editor-in-Chief

PLOS Genetics

**Journal Requirements:**

- Please ensure that the CRediT author contributions listed for every co-author are completed accurately and in full.

At this stage, the following Authors/Authors require contributions: Melissa M. Rolls. Please ensure that the full contributions of each author are acknowledged in the "Add/Edit/Remove Authors" section of our submission form.

The list of CRediT author contributions may be found here: https://journals.plos.org/plosgenetics/s/authorship#loc-author-contributions

**Reviewers' comments:**

Reviewer's Responses to Questions

**Comments to the Authors:**

Reviewer #1: In this manuscript entitled “DLK orchestrates a modular transcriptional response to axon injury with separate roles for Fos and Jun” by Mahata and colleagues, the transcriptional network downstream of DLK in the context of axon regeneration is explored. The authors use Drosophila larvae, in which Class I ddaE sensory neurons are axotomized to study the regenerative capacity. They employ the Gal4/UAS system to cell-autonomously manipulate various candidates to test their requirement for regeneration and the temporal sequence in which these candidates enter or exit the nucleus. Drosophila is an excellent model system for dissecting the regenerative capacity of sensory neurons, and gaining insight into the underlying mechanisms may provide novel insights for developing therapeutic approaches in humans.

In the current manuscript, the authors convey the following observations:

- Fos, Jun, STAT, Atf3, and Slbo are required for axon regeneration of sensory neurons.

- Fos controls early injury responses independently of Jun, using puc-GFP as a reporter of early responses.

- An increase of nuclear Fos following injury due to Fos homodimerization, while Fos/Jun hetero- and Jun/Jun homo-dimerizations are reduced.

- DLK/Jun activation disrupts Jun homodimerization.

- Nuclear STAT accumulation is negatively regulated by Fos, but requires Jun.

- The nuclear dynamic of Atf3 is regulated by DLK, Fox, Jun, and STAT.

This manuscript would be suitable for the general readership of PloS Genetics. But in its current form, it requires extensive polishing and reworking. I have provided several major and minor comments below, which aim to improve the quality of the manuscript to ensure it is suitable for publication. They should be adressed.

Major comments (A-F):

A) In Figure 3, the authors generated a Fos-mNG reporter. It is important to perform a proof-of-concept experiment, where the nuclear Fos localization should be dependent on DLK.

B) Lines 178-81: Quote “Thus, the five transcription factors most closely linked to DLK signaling after axon injury across multiple systems are all required for axon regeneration in Drosophila making it a useful system to understand their relationships. We chose to pursue the four transcription factors with the strongest phenotype for additional experiments.”

It would be useful for the reader if the authors mention which one of the transcription factors has the weakest phenotype and is left out for subsequent studies.

C) The authors state in the text (line 335), quote: “In summary, our data suggests that Jun phosphorylation after axon injury disrupts its homodimerization, likely allowing it to partner with another bZip transcription factor.” And they phrased the legend of Figure 5 (line 1042) as follows: “DLK/JNK signaling pathway promotes Jun homodimer dissociation through phosphorylation.”

However, the effect is rather mild, and the only experiment that was perfomed used the phosphomimetic Jun variant. Further experimental support should be provided, or the conclusion in the text and the title in the figure legend should be toned down.

D) Figure 3: Why do the authors explain the knock-in schematic at the end of Figure 3, in the display item 3E? It would make more sense to provide the concept in display Item 3A and show the data in 3B-E.

E) Lines 341-90: In these paragraphs, the authors reference Figure 6. First, they reference 6A, B, then 6C, D, then 6I, J, then back to 6E, F, and finally 6G, H. Why aren’t the authors adjusting the figures according to the flow of the text? Jumping back and forth between figures while reading the text adds an unnecessary level of complexity.

F) Figure 6: In the display items C and D, the examples do not match the quantification. Please correct. In the display item K, the inhibitor of Fos should be as big as the arrows of all the mediating proteins.

Minor comments (1-7):

1) Figure 1C: The description of the genotypes is not clear in the figure. What is the difference between the two Jun RNAi experiments (Jun v. Jun mut; Jun RNAi)? The authors should clarify this better in the text and the figure legend.

2) Figure 1D: To improve the clarity for non-experts, it would be good to indicate the regenerating axon (that fails to grow out), as done for the control with the purple dashed line.

3) Line 177: The authors mentioned AR, but did not explain what the abbreviation stands for. I assume it means axon regeneration. I suggest writing it out and putting AR in parentheses.

4) Figure 7: In the display items A and B, the 6 hpi example is left out, but included in the quantification. Please correct.

5) Figure S1: The genotypes are not consistent between the examples and the quantification. Also, is the Atf3 over-expression referenced in the text? It would be better to separate the Atf3 over-expression from the Jun overexpression data, either in a different supplementary figure, or in a separate display item.

6) Figure S2: In its current form, the figure is hard to read because the examples and quantification do not match. It would be beneficial to split the display items further and provide quantification for each example. I also suggest adding insets, as done in all other figures.

7) Lines 983, 117, 1142, 1153, and 1164: The authors mentioned error bars in the figure legend, but the figure does not contain error bars. Do they mean the dashed lines? If so, please explain it better.

Reviewer #2: In this article, the authors aimed to dissect out the roles of different bZip transcription factors which are individually described in the context of axonal injury response. Fos, Jun, ATF etc. It was known that DLK acts upstream of these transcription factors. However, it was unclear if they act independently or as part of a connected sequence. The authors show that DLK-regulated ranscription factors- Fos, Jun, STAT and Atf3 are required for axon regeneration in Drosophila model, but act during different phases. They suggest that Fos is required for early axon injury responses that stabilizes the cell. Jun is activated later by DLK, independent of Fos and it regulates STAT and Atf3. This second module is needed for later injury responses including axon regeneration. This work provides interesting insights with genetically encoded fluorescent reporters of these Transcription factors. It clearly shows FOS and Jun are working in different time windows. The manuscript has a potential to be published in PLOS Genetics. I suggests the authors to address the following concerns

1) In Figure-1, the authors showed that FOS TF is required for axon regrowth using EBP-GFP microtubule reporter. Same phenotype was noticed due to loss of jun and few other TFs. However, author proposes FOS and JUN is working in two phases in axon regeneration. In the Figure-2D, the author showed that the overexpression of FOS also reduced the axon regrowth like the loss of fos. How would the opposing genetic manipulation cause same phenotype?

2) The authors suggest that specially, FOS might be acting in early neuroprotection phase. I did not find any neuroprotection assay in this manuscript. Will there be higher frequency of degeneration for the injured neuron in the absence of fos ? and opposing effect due to overexpression of FOS?

3) In figure-2: It will be nice if authors describe the puc::GFP reporter assay. How it is an established reporter for injury response?

Is the phenotype of loss of puc is like the loss of fos or dlk?

4) In Figure-4, What would be the effect of DLK knockdown on Fos homodimerization?

5) In this report, the authors use the repolarization as an assay (injury done adjacent to cell body) for axon regrowth. what happens if the cut (axotomy) is done in a distal position in the axon? Whether axon regrows if the injury is done in the distal portion?

In that paradigm whether the knockdown of fos and jun will perturb axon regrowth to similar extent?

Reviewer #3: The current manuscript from the Rolls lab addresses genetic and temporal relationships for transcription factors in the axon injury response in Drosophila sensory neurons. The question is important because many transcription factors have been shown to be induced by axon injury, but their functional roles, and relationships to each other have not been clear. The authors chose to compare the functions and subcellular localization of a handful of these conserved in axon injury in the Drosophila sensory neuron ddaE. The authors find that knockdown of Fos, Jun, ATF3, and STAT, all reduce axon regenerative capacity following axon injury. They then go on to show both differential regulation of these proteins by the DLK stress sensor as well as different regulatory relationships with each other. The questions addressed here are important and the authors made some new reagents: Fos-mNG, Jun-mNG and BiFC Fos that will be useful to the field. The major finding that Fos and Jun likely have different roles in the injury response is reasonably well supported. This paper would be clearer if it were tightened up around the central claims regarding Jun and Fos activity in axon regeneration. The data separating their roles in the regulation of STAT and Atf3 is confusing and detracts from the major claim.

Several major concerns undercut my enthusiasm for a potentially important study. (1) The use of single RNAis for functional analysis. In two cases (Fos and Jun), the RNAis were put in a heterozygous background for some (not all) assays, and the phenotypes were modified, emphasizing the fact that the RNAis give only partial knockdown. This really undercuts confidence in the results—especially when the goal of the experiences (to order a pathway of genes with quite similar phenotypes) very tricky. (2) The authors seem to assume that Jun acts following injury, presumably based on phospho-Jun increases following injury in other systems. However, in this system, it doesn’t seem clear that Jun is required following injury. Is it possible it is required developmentally and establishes competence of the neuron to respond? (3) The data that Jun controls Atf3 activation while Fos does not is not at all convincing. (4) I am not convinced that Atf3 regulation makes sense in the pathway as presented since it is a transcription factor that presumably acts in the nucleus; yet, as proposed the pathway in this manuscript regulate its export.

More detailed comments:

• they should use more than 1 RNAi line per gene. Or at a minimum, they need to provide validation of the single RNAi lines they used to ascertain function. They establish that Jun RNAi line they use does not strongly knockdown Jun levels because the regeneration phenotype is strongly enhanced in a Jun heterozygote. They similarly put the Fos RNAi into a het background for some analyses. But these background were not always used, and similar approaches were not taked for Atf3 and STAT.

• The authors propose that Fos is necessary for “early step of the axon injury response”, which they think is neuroprotection. This conclusion is based on the requirement for Fos in Puc induction and in the confusing ability of Fos OE to suppress regeneration. The data that Fos is required specifically for the early injury response versus DLK-dependent response is weak.

• Potentially useful new reagents to track subcellular localization of Fos and Jun (Fos-mNG and Jun-mNG). They use them to report on different time courses of Fos and Jun following injury. Before using them as reporters of Fos and Jun subcellular localization, they need to first evaluate whether the endogenous tags have impaired function of either Fos or Jun by testing whether axon regeneration is impaired in a homozygous tag background. Otherwise, it is not clear whether they are faithful reporters.

• Jun is clearly required for axon regeneration, but its nuclear levels do not increase following injury. Is it possible that the temporal requirement for Jun is before injury such that Jun is required to establish competence for the neuron to regenerate its axon.

• The comparison between the ability of wildtype and phosphomimetic Jun to reduce fluorescence from the BiFC-fragment-tagged Jun is uninterpretable unless the transgenes are inserted at the same genomic location. They should also test whether the reduction in BiFC signal is simply due to Gal4 dilution by testing the degree to which it is reduced via the introduction of another UAS transgene.

• The idea that Fos regulates DLK signaling and Jun does not is clear. On the other hand, the authors should soften their claims that Fos and Jun regulate axon injury at different time points, as they are inferring this based on changes in nuclear localization. This is particularly problematic in the case of Jun where it is unclear what nuclear localization corresponds to functionally.

• The model that STAT is controlled by Jun is not convincing. Its nuclear localization is only modestly affected by Jun, and this effect seems suppressed when the RNAi is put into a jun heterozygous background. These authors should comment on this. Additionally, there is not a meaningful difference in the nuclear STAT localization in jun knockdown in the het background versus fos knockdown. Thus, it is unclear why the authors assert that STAT is downstream of Jun and not Fos.

• Inclusion of Atf3 in a unified pathway is confusing since Atf3 RNAi has the same effect on axon regeneration as the other genes studied in this ms, but DLK, Jun, Fos, and STAT all appear to negatively regulate its expression since Atf3 nuclear localization is increased when these genes are knocked down. As presented, these data do not appear consistent.

**Have all data underlying the figures and results presented in the manuscript been provided?**

Reviewer #1: Yes

Reviewer #2: Yes

Reviewer #3: Yes

PLOS authors have the option to publish the peer review history of their article (what does this mean? ). If published, this will include your full peer review and any attached files.

**Do you want your identity to be public for this peer review?** For information about this choice, including consent withdrawal, please see our Privacy Policy .

Reviewer #1: No

Reviewer #2: No

Reviewer #3: No

**Figure resubmission:**
---

## [Decision Letter · Decision Letter 1]

24 Nov 2025

Dear Dr Rolls,

We are pleased to inform you that your manuscript entitled "DLK orchestrates a modular transcriptional response to axon injury with separate roles for Fos and Jun" has been editorially accepted for publication in PLOS Genetics. Congratulations!

Yours sincerely,

Hongyan Wang, Ph.D.

Academic Editor

PLOS Genetics

Fengwei Yu

Section Editor

PLOS Genetics

Aimée Dudley

Editor-in-Chief

PLOS Genetics

Anne Goriely

Editor-in-Chief

PLOS Genetics

BlueSky: @plos.bsky.social

Comments from the reviewers (if applicable):

Reviewer's Responses to Questions

**Comments to the Authors:**

Reviewer #1: None.

Reviewer #2: The authors have addressed all the concerns. The revised manuscript significantly increased the clarity of the findings and the model the authors are conveying.

Therefore, it is suitable for publication in PLOS-Genetics.

**Have all data underlying the figures and results presented in the manuscript been provided?**

Reviewer #1: Yes

Reviewer #2: Yes

PLOS authors have the option to publish the peer review history of their article (what does this mean? ). If published, this will include your full peer review and any attached files.

**Do you want your identity to be public for this peer review?** For information about this choice, including consent withdrawal, please see our Privacy Policy .

Reviewer #1: **Yes: ** Lukas Neukomm

Reviewer #2: No

**Data Deposition**

http://datadryad.org/submit?journalID=pgenetics&manu=PGENETICS-D-25-00607R1

**Press Queries**

---

## [Editor Report · Acceptance letter]

PGENETICS-D-25-00607R1

DLK orchestrates a modular transcriptional response to axon injury with separate roles for Fos and Jun

Dear Dr Rolls,

We are pleased to inform you that your manuscript entitled "DLK orchestrates a modular transcriptional response to axon injury with separate roles for Fos and Jun" has been formally accepted for publication in PLOS Genetics! Your manuscript is now with our production department and you will be notified of the publication date in due course.

With kind regards,

Anita Estes

PLOS Genetics

On behalf of:
